# Bifidobacteria shape antimicrobial T-helper cell responses during infancy and adulthood

Katrin Vogel [1], Aditya Arra [1], Holger Lingel [1], Dirk Bretschneider[2], Florian Prätsch [3], Denny Schanze [4], Martin Zenker [4], Silke Balk[1], Dunja Bruder[5,6], Robert Geffers[7], Thomas Hachenberg[3], Christoph Arens [8,9] & Monika C. Brunner-Weinzierl [1] ✉

Microbial infections early in life are challenging for the unexperienced immune system. The SARS-CoV-2 pandemic again has highlighted that neonatal, infant, child, and adult T-helper(Th)-cells respond differently to infections, and requires further understanding. This study investigates antibacterial T-cell responses against *Staphylococcus aureus aureus*, *Staphylococcus epidermidis* and *Bifidobacterium longum infantis* in early stages of life and adults and shows age and pathogen-dependent mechanisms. Beside activation-induced clustering, T-cells stimulated with *Staphylococci* become Th1-type cells; however, this differentiation is mitigated in *Bifidobacterium*-stimulated T-cells. Strikingly, prestimulation of T-cells with Bifidobacterium suppresses the activation of *Staphylococcus*-specific T-helper cells in a cell-cell dependent manner by inducing FoxP3⁺CD4⁺ T-cells, increasing IL-10 and galectin-1 secretion and showing a CTLA-4-dependent inhibitory capacity. Furthermore *Bifidobacterium* dampens Th responses of severely ill COVID-19 patients likely contributing to resolution of harmful overreactions of the immune system. Targeted, age-specific interventions may enhance infection defence, and specific immune features may have potential cross-age utilization.

A child faces the greatest probability of dying within the first 28 days after birth, followed by the first 5 years of life[1,2]. Although studies have shown that the immune system early in life reacts differently compared to that of adults, the developmental-specific mechanisms of triggering an antibacterial response and avoiding an adverse reaction are yet not completely understood[3–7]. Recently, a predetermined developmental trajectory of the composition of the cell populations in blood during the first 3 months of life was reported; however, data thereafter and functional characteristics of these cells are not addressed in depth[8,9]. In this respect, T-helper(Th)-cells from neonates and infants[10–12] were shown to generate fungal-specific primary T-cell responses that differ between infancy and adulthood, with neonatal-characteristic T-cells showing up in the adult memory pool[4,7].

[1]Department of Experimental Paediatrics, University Hospital, Otto-von-Guericke University, Magdeburg, Germany. [2]Department of Paediatrics, Hospital St Marienstift, Magdeburg, Germany. [3]Department of Anaesthesiology and Intensive Care Medicine, University Hospital, Otto-von-Guericke-University, Magdeburg, Germany. [4]Institute of Human Genetics, University Hospital, Otto-von-Guericke University, Magdeburg, Germany. [5]Infection Immunology Group, Institute of Medical Microbiology and Hospital Hygiene, Health Campus Immunology, Infectiology and Inflammation, Otto-von-Guericke University, Magdeburg, Germany. [6]Immune Regulation Group, Helmholtz Centre for Infection Research, Braunschweig, Germany. [7]Genome Analytics, Helmholtz Centre for Infection Research, Braunschweig, Germany. [8]Department of Otorhinolaryngology, Head and Neck Surgery, University Hospital, Otto-von-Guericke University, Magdeburg, Germany. [9]Present address: Justus-Liebig-University Gießen, University Hospital of Gießen and Marburg (UKGM), Gießen Campus, Department of Otorhinolaryngology, Head/Neck Surgery and Plastic Surgery, Gießen, Germany. ✉e-mail: Monika.Brunner-Weinzierl@med.ovgu.de

*Bifidobacterium longum ssp. infantis* (*B. infantis*) is capable of rapidly colonizing the infant gut early in life, and its growth is supported by components found in human milk[13–15]. It outcompetes *Staphylococcus* and *Streptococcus* and decreases inflammation[16,17]. *B. infantis* has been shown to reduce susceptibility to diseases that are driven by unwanted T-cell responses; for instance, in mouse models of atopic airway disease, *B. infantis* has been shown to solve pathology correlated with increased numbers of regulatory T-cells (Treg)[18]. Beneficial Treg effects mediated by *B. infantis* have also been demonstrated to dampen the side effects of immune checkpoint therapies in a tumor model[19,20]. In contrast, *S. aureus* is the most common pathogen of skin and soft tissue infections responsible for a broad range of clinical manifestations[21] and is one of the most common organisms isolated from children with healthcare-associated infections[22,23]. *S. aureus* can cause excessive T-cell activation and dysregulated inflammation, resulting in the release of high levels of cytokines (IFNs, ILs, chemokines, CSFs, TNF, etc.) into the bloodstream, which has a systemic and detrimental effect on multiple organs[24,25]. Although it is well recognized that the immune system of neonates and infants differs from that of adults, not much is known about its ability to specifically respond to bacterial antigens such as those of *Staphylococci* and *Bifidobacterium*[26]. These bacteria are of particular interest because newborns are immediately and continuously exposed to them throughout their lifetime. Therefore, it is expected that there has been a co-evolution of this human-microbe interaction that has shaped the human adaptive immune response, especially early in life. We hypothesize that antigen-specific pro- and anti-inflammatory T-cell responses to abundant bacteria are formed very early in life, probably in an age-specific manner.

Here, we investigated the cellular mechanisms of CD4$^+$ T-cell differentiation triggered in response to *S. aureus, S. epidermidis*, or *B. infantis* and analyzed the impact of their interplay on each other. We show that CD4$^+$ T-cells show a response characteristic of their age as well as pathogen-specific. From a mechanistic point of view, we found that *B. infantis* particularly promotes regulatory T-cell responses by up-regulation of genes characteristic for regulatory T-cells such as FoxP3, GITR, IL-10, and CTLA-4. While enhanced FoxP3$^+$ Treg formation, galectin-1 release, IL-10 secretion, as well as suppressing severe effector responses by *B. infantis* stimulated cells take place in a cell-cell contact and CTLA-4-dependent manner.

## Results

### Neonatal, infant, and children's T-cells recognize and respond to *S. aureus* and *B. longum ssp. infantis*

To determine the characteristics of antigen-specific CD4$^+$ T-cell differentiation against bacteria from birth to childhood, we have set up an ex-vivo model using naïve CD4$^+$ T-cells of donors aged 0–12 years (Tables S1 and S2) and adults (Table S3). Naïve CD4$^+$CD45RA$^+$CD31$^+$ (recent thymic emigrants) T-cells routinely enriched to a purity of >99.6% from cord blood (CB) (example for the youngest T-cells), blood, and adenoids of infants and children and from peripheral blood (PB) of adults were antigen-specifically stimulated and analyzed in this study. The antigen presentation to T-cells was mediated by autologous CD14$^+$ monocytes that were matured 16 h with heat-inactivated (h.i.) *B. longum ssp. infantis (B. infantis)*, h.i. *S. aureus ssp. aureus, or* h.i. *S. epidermidis*, respectively (see "Materials and methods"). Similar maturation between age groups analyzed was controlled by surface expression of CD16 and HLA-DR as well as intracellular expression and accumulation of secreted cytokines IL-6 and IL-1ß (Fig. S1a, b; Tables S43–47)[7]. Initial monitoring of TCR-induced upregulation of activation-associated marker CD25 shows that - although at different frequencies - T-cells from donors of all age groups analyzed were activated against *S. aureus, S. epidermidis*, and *B. infantis* after 3 days of stimulation (Fig. 1a, Fig. S2a, b). Using non-specific anti-CD3/CD28-mediated activation as a positive control, T-cells show strong

activation at any age (Fig. 1a, black line). Subsequently, activation-induced cell clusters were fluorescently measured and analyzed by an Immunospot reader that revealed unambiguously >100 clusters per 10$^5$ cells in samples of both neonates and adults (Fig. 1b, c, Table S5). Resting T-cells did not form any clusters, whereas *S. aureus*, as well as *B. infantis* stimulation, led to large-scale cell clustering, implicating that bacterial antigens are well recognized by T-cells. Different effects of cell death triggered by h.i. *S. aureus*, h.i. *S. epidermidis* as well as h.i. *B. infantis* and different age groups could be excluded (Fig. S2c, Table S48).

Proliferation in response to antigen-specific stimulation is an essential feature of T-cells during an immune reaction. Therefore, the activation-induced proliferation of T-cells was determined by labeling cells with a vital dye carboxyfluorescein succinimidyl ester (CFSE) prior to 3-day stimulation with bacteria-matured APCs (see above). T-cell proliferation was unambiguously detectable by CFSE fluorescence dilution upon stimulation with h.i. *S. aureus*, h.i. *S. epidermidis* and also h.i. *B. infantis*. Importantly, blockage of HLA-DR prevented activation-induced T-cell proliferation confirming antigen-specificity (Fig. 1d, e, Fig. S2d, e, Tables 49 and 50), whereas T-cells at any age proliferated vigorously upon non-specific anti-CD3/CD28 mediated activation (positive control) (Fig. 1e black line). In response to all bacteria analyzed, *S. aureus* induced the strongest proliferative response, followed by *S. epidermidis* and *B. infantis*; the capacity to proliferate was always profoundly increased in T-cells from the younger donors. Correlating the bacterial antigen-mediated proliferation of T-cells with the age of donors revealed pathogen-specificity and demonstrated age dependency, as shown by Pearson's correlation coefficient ($r < -0.67$, all $p$-values $< 0.001$) and Spearman's Rho (Rho $< -0.865$, all $p$-values $< 0.001$) (see Fig. 1e). In detail, *S. aureus*-specific stimulation showed higher frequencies of proliferated T-cells in neonates and infants up to 2 years (37.5% and 40%), compared to children aged 6–12 years (18.4%) or adults (18.9%) (Fig. 1d, e, Fig. S2c, d). T-cell proliferation induced by *B. infantis* was detectable only until the age of two, whereas *S. aureus* and *S. epidermidis* significantly triggered a proliferation of naïve T-cells at any age examined (Fig. 1d, e, Fig. S2d, e). These results reveal that the capacity of naïve T-cells to specifically recognize and respond to bacterial antigens of *S. aureus, S. epidermidis*, or *B. infantis* is pathogen and age-dependent.

### Age-dependent expression and secretion of cytokines

To determine whether T-cell antigen-specific responses to *S. aureus* and *B. infantis* at different stages of life have divergent differentiation characteristics, type-1 and type-17 signatory cytokine expression and secretion were monitored. Therefore, T-cells from neonates, children, and adults were stimulated for 3 days with h.i. *S. aureus* or h.i. *B. infantis* matured monocytes as described in Fig. 1, and the intracellular expression and supernatant accumulation of cytokines were determined by flow cytometry and cytokine capture assay, respectively. The expression of the Th1 master cytokine IFN-γ (Fig. 2a, b, right panel, Table S6), as well as the Th1-associated master transcription factor T-bet (Fig. 2c, Table S7) was detectable in all *S. aureus* stimulated T-cells. In extension to a published study of total CD4$^+$ T-cells[3], experiments were performed using naïve T-cells. The accumulation/secretion of IFN-γ was three times higher in supernatants of *S. aureus*-simulated T-cells of donors being 6 years and older compared to 0–5-year-olds (Fig. 2b, right) ($p < 0.0001$). In *B. infantis* stimulated T-cells, IFN-γ producers were detectable at low frequencies but significantly higher in children and adults in comparison to neonates. In contrast, even though at low frequencies, *B. infantis* stimulated, T-cells had more IL-2 producers in neonates, infants, and children than in adults (Fig. 2a, b, left). Additionally, expression of TNF-α was monitored in neonatal T-cells in comparison to all other age groups (Fig. 2a, middle). Indeed, higher frequencies of TNF-α producing neonatal T-cells were

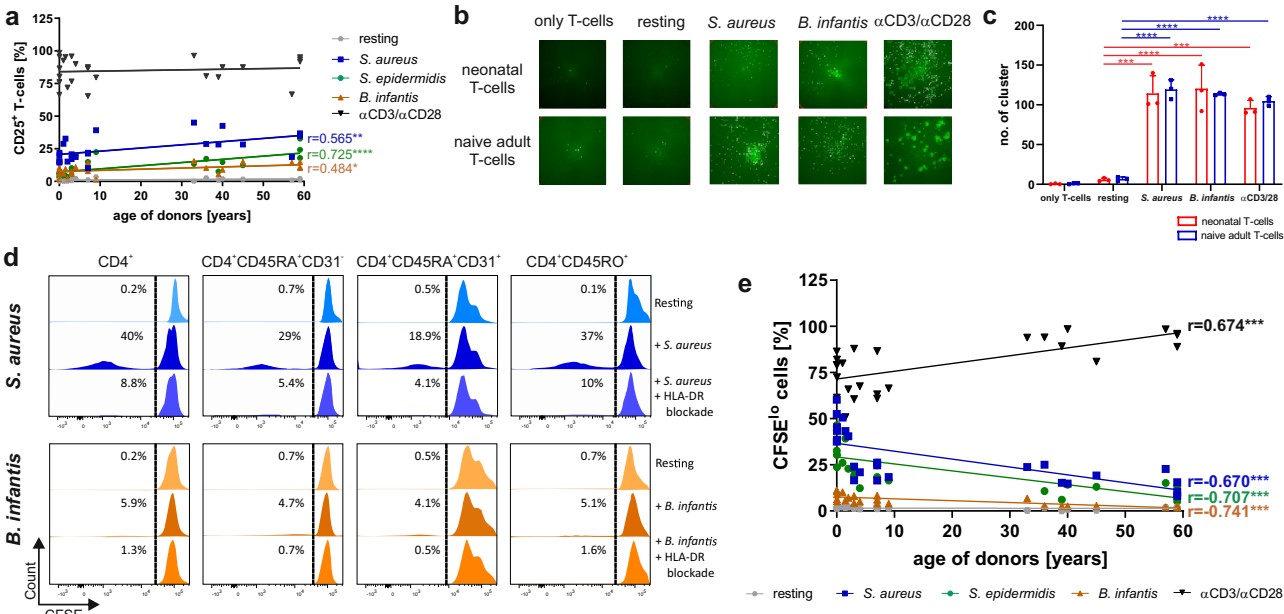

**Fig. 1 | Bacteria-specific T-cell activation. a** After 3-day incubation of monocytes previously matured with h.i. *S. aureus* (blue), h.i. *S. epidermidis* (green), or h.i. *B. infantis* (orange) and naïve T-cells or naïve T-cells stimulated with anti-CD3/CD28 (black), the frequency of CD25-expressing CD4⁺ T-cells from neonates, infants, children, and adults was measured by flow cytometry and plotted against age. **b** Purified naïve (CD4⁺CD45RA⁺CD31⁺) T-cells from neonates and adults were labeled with CSFE and co-cultured for 3 days with monocytes matured with h.i. *S. aureus* or h.i. *B. infantis* or anti-CD3/CD28 as indicated. CFSE-labeled cells were counted using the ImmunoSpot S6 ULTIMATE UV Image Analyzer to determine the number of cell clusters formed after stimulation. **c** Bar graphs show the number of clusters formed by three different donors. Each dot represents a different donor. Error bars in the figures denote mean + SD. *$p < 0.05$, **$p < 0.01$, ***$p < 0.001$, ****$p < 0.0001$ as determined by one-way ANOVA with Holm–Sidak post hoc test. Detailed statistical data are provided in Supplementary Table 5.

**d** The indicated T-cell subsets of a representative adult donor were co-cultured with monocytes matured with h.i. *S. aureus*-antigen or h.i. *B. infantis*-antigen in the presence or absence of HLA-DR blocking antibody. Flow cytometry measured the proliferation of CD4⁺, CD4⁺CD45RA⁺, CD4⁺CD45RO⁺, and CD4⁺CD45RA⁺CD31⁺ T-cells after 3 days of stimulation. Data are representative of five donors. **e** Frequency of proliferating (CFSE^lo) cells of T-cells from neonates, infants, children, and adults that were stimulated for 3 days with h.i. *S. aureus* (blue), h.i. *S. epidermidis* (green), h.i. *B. infantis* (orange) or anti-CD3/CD28 (black), determined by flow cytometry and plotted against donors age. Cumulative results are shown, and each dot represents one donor (**a, c, e**). Correlation analyses were performed using Pearson's correlation. Pearson's correlation coefficient r is shown with *** representing $p < 0.001$. All data shown are representative of at least three experiments performed with T-cells from different donors. Source data are provided as a Source Data file.

observed in response to *S. aureus* or *B. infantis* (Fig. 2a, b), compared to T-cells of children or adults. In addition, enhanced concentrations of secreted TNF-α were monitored in samples of neonates in response to all bacteria analyzed (Fig. 2b). Considering IFN-γ producers for the representation of Th1-like T-cells, the age-differential propensity to elicit Th1 responses was additionally determined, as shown for *S. aureus* (Fig. 2a, b) and for *S. epidermis* (Fig. 2d, Table S8).

Of note, potential bacterial triggers of the innate immune system, such as signaling of MyD88-dependent TLR receptors or complement receptor 3α to initiate the responses, were determined to be neglectable using specific inhibitors or blocking antibodies (Fig. 2e, Fig. S3, Table S9, S51–54). Next, T-cells, which express several cytokines simultaneously and are thus considered particularly potent effector T-cells, were examined after bacteria-specific stimulation[27]. Only in response to *S. aureus*, Th1-like cytokine-co-producers could be detected in substantial frequencies (Fig. 2f, Table S10), but not in response to *B. infantis* or *S. epidermidis*, which mainly generated cytokine single-producers (Fig. S4; Table S55 and 56). Analysis of cytokine co-producing T-cells showed that only neonates, infants, and children up to 2 years of age developed significantly more double and triple producers following 3 days of stimulation with *S. aureus* (Fig. 2f).

As studies utilizing non-specific T-cell responses have identified IL-17 as a signature cytokine for neonatal T-cells[28], we investigated its expression along with the Th17 master transcription factor RORγt in bacterial-specific activated primary T-cells (Fig. 2g, h, Tables S11–S13). The highest frequencies of IL-17A⁺ *S. aureus*- and *S. epidermidis*-specific T-cells were documented in neonates and infants when compared to

children and adults (Fig. 2g, left bar graph). A similar tendency was associated with the IL-17 master transcription factor RORγt, as neonatal T-cells displayed 4-times enhanced expression of RORγt compared to T-cells of adults (Fig. 2h). These data highlight that the antigen-specific T-cell differentiation shows a bias to IL-17 signature cytokine production (Th17-like cells) early in life, plus that it perpetuates to some extent during infancy and childhood. Indeed, the type of pathogen determines at what age this Th17-bias is thwarted, as is for the *S. aureus*-specific responses after 5 years of age, for *S. epidermidis*-specific responses before 0.5 years of age and for *B. infantis*-specific responses after 12 years of age (Fig. 2g, left and right bar graphs).

## *B. infantis*-mediated T-cell differentiation is CTLA-4 dependent and involves the generation of FOXP3⁺ T-cells

Next, we investigated the mechanism behind *B. infantis*-specific T-cell activation as determined mainly by abundant cluster formation (Fig. 1). Though low (still about 8-10% responding human CD4⁺ T-cells in an antigen-specific, polyclonal setting), but significantly positive, response in terms of frequencies of CD25-expressing cells and proliferation or cytokine production (Figs. 1 and 2). Therefore, we isolated CD4⁺CD45RA⁺CD31⁺ T-cells from donors aged 0–12 and adults (Fig. 3a) and stimulated them with *B. infantis*-pulsed APCs for 3 days as described in Fig. 1 in the presence and absence of blocking antibodies specific for HLA-DR. Activation-induced TCR-mediated upregulation of CD25 was prevented significantly by applying HLA-DR-blockade, confirming the antigen-specificity of the response (Fig. 3a, Table S14).

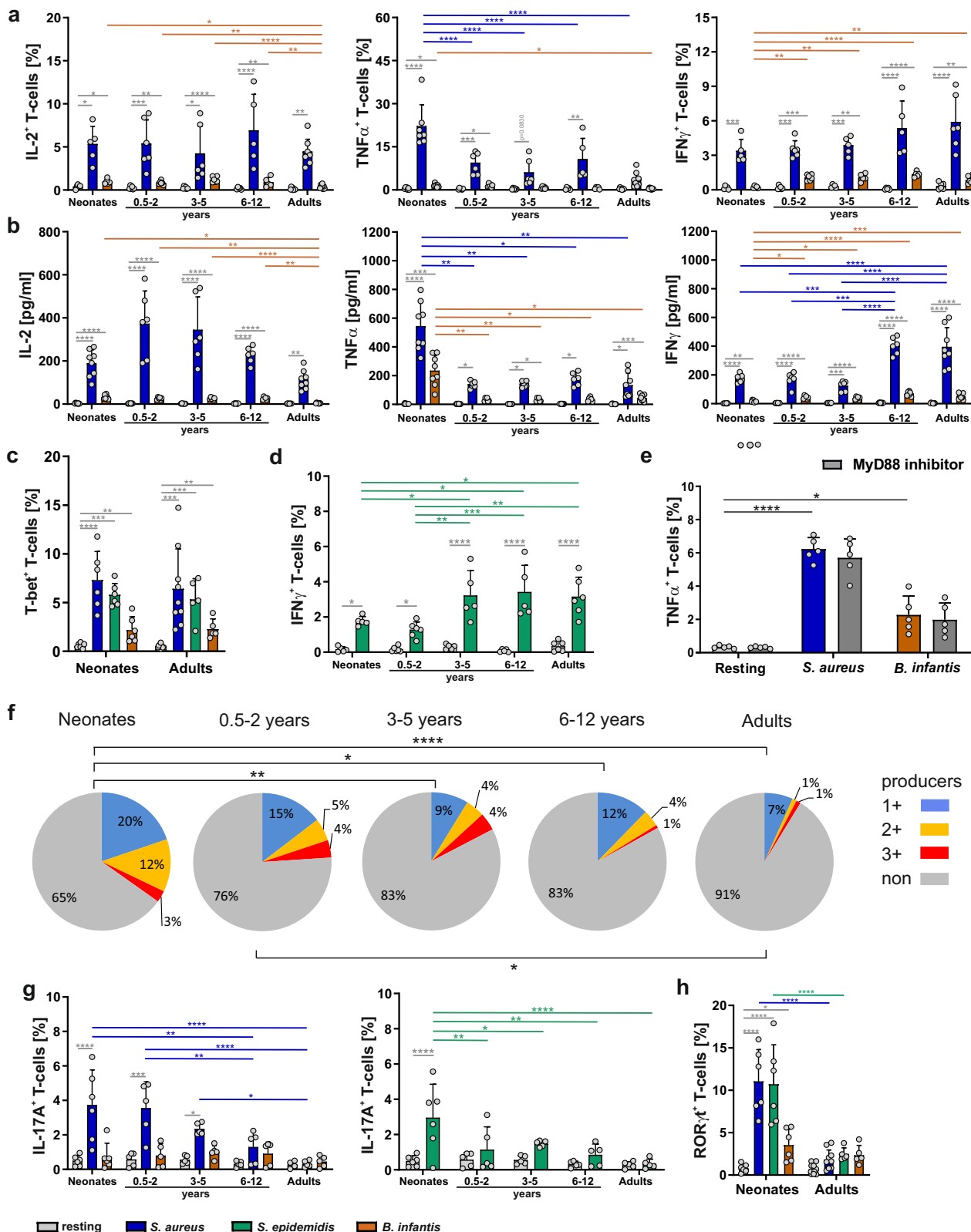

To gain a deeper understanding of the pathways specifically induced by *B. infantis* in T-cells, we performed RNA sequencing assays on these cells. Moreover, since it has already been shown that *Bifidobacterium* modulates metabolic processes and increases mitochondrial activity in intestinal Tregs[20], we focused our analysis on genes that activate molecules that are distinctive to Treg cells. After a literature search, we compiled a list of 18 genes for molecules whose up- or downregulation is characteristic of Treg cells or known for inhibitory functions (Table S15)[17,29–31], including co-stimulatory or co-inhibitory surface molecules of Th and Treg cells, and extracted them from the RNASeq data (Fig. 3b, Fig. S5). In neonates, 11 of the 18 listed genes for the molecules were enriched or decreased by more than twofold compared with resting T-cells, and 9 out of them show even a high significance level of FDR < 0.05. We found increased gene transcripts

**Fig. 2 | Bacteria-specific cytokine expression by T-cells from different age groups. a** Frequency of T-cells from neonates, infants, children, and adults expressing intracellular IL-2, TNF-α, or IFN-γ after stimulation with *S. aureus* (blue), *S. epidermidis* (green), or *B. infantis* (orange) matured monocytes for 3 days was determined by flow cytometry. **b** Determination of IL-2, TNF-α, or IFN-γ cytokine release from CD4⁺CD45RA⁺CD31⁺ T-cells, which were either stimulated or resting for 3 days. **c** T-cells from neonates and adults stimulated with *S. aureus, S. epidermidis,* or *B. infantis* matured monocytes for 3 days. Expression of transcription factor T-bet was determined by flow cytometry. **d** Frequency of T-cells stimulated with *S. epidermidis* matured monocytes expressing intracellular IFN-γ. **e** After MyD88-inactivation during adult CD4⁺CD45RA⁺CD31⁺ T-cells stimulation against *S. aureus* or *B. infantis* using the specific inhibitor Pepinh-MYD, intracellular cytokine expression of TNF-α was determined by flow cytometry and the frequency of these cells is shown as a bar graph. **f** CD4⁺CD45RA⁺CD31⁺ T-cells were stimulated with *S. aureus,* and the cells expressing single or multiple cytokines IL-2, TNF-α, and IFN-γ were determined by flow cytometry, analyzed by Boolean gating and shown as a

fraction of all CD4⁺ T-cells in a pie chart. The subsets that simultaneously express no (gray), one (blue), two (yellow), or three (red) different cytokines are grouped by color. Data are representative of five donors. *p < 0.05, **p < 0.01, ***p < 0.001, ****p < 0.0001 as determined by Fisher exact test. **g** Bar graphs showing the frequency of CD4⁺CD45RA⁺CD31⁺ T-cells expressing intracellular IL-17A after 3 days of stimulation with either *S. aureus, B. infantis* (left bar graph), or *S. epidermidis* (right bar graph) matured monocytes. **h** Neonatal and adult CD4⁺CD45RA⁺CD31⁺ T-cells were stimulated with *S. aureus, S. epidermidis,* or *B. infantis* matured monocytes for 3 days, and the cells expressing the transcription factor RORγt were determined by flow cytometry. Error bars in the figures (**a**–**e**) and (**g**, **h**) indicate mean + SD, *n* ≥ 5 donors from at least 3 independent experiments in each age group, *p*-values in (**a**–**e**) and (**g**, **h**) are calculated by Kruskal Wallis test corrected by Dunn's post hoc test, *p < 0.05, **p < 0.01, ***p < 0.001, ****p < 0.0001. Detailed statistical data are provided in Supplementary Tables 6–13. Source data are provided as a Source Data file.

of signatory molecules for Treg cells, such as FoxP3 (IPEX), CTLA-4 (CD152), Helios (IKZF2), CD25 (IL2RA), GITR (TNFRSF18), and genes mediating inhibitory functions, such as PD-1 (PDCD-1, CD279), the cytokines IL-10 (*p* = 0,019, FDR = 0,07) and IL-35 (EBI3), as well as CD103, whereas LAG3 (CD223) and TGFβ (TGFB1) were not regulated. Of note, the RNA profile of the metabolite galectin-1 (LGAL-S1) was increased in neonates with high fold change but not significantly different (FDR = 0.24), whereas a significant change of the same (FDR = 0.0107) was observed in *B. infantis*-stimulated adult T-cells (Fig. S5). Furthermore, we identified gene products downregulated in *B. infantis*-stimulated neonatal T-cells, such as 4-1BB (TNFSF9, CD137), a costimulatory activation-associated molecule of effector Th cells; this down-regulation is a signatory for Treg cells. In addition, we conducted a Gene Ontology (GO) enrichment analysis of the RNASeq data (Fig. S6). Since our findings point toward Treg cells, we hypothesized that these cells would exhibit their characteristic metabolism[32,33]. By investigating the metabolic profile of neonatal T-cells responding to *B. infantis*, we discovered that T-cells stimulated by *B. infantis* indeed demonstrated significantly increased oxidative metabolism, including oxidative phosphorylation (OXPHOS) that is typical of Treg cells and contributes to maintaining their Treg phenotype and suppressive capacity (Fig. S6)[32,33].

Although the RNASeq data showed a moderate increase in the expression of galectin-1 mRNA in neonatal T-cells, it was stronger in adult T-cells. As galectin-1, a member of the family of beta-galactoside-binding proteins that has been demonstrated to be overexpressed in inducible Treg cells[34], mediated its suppressive functions and was suggested in mouse tissue for being relevant for *B. infantis*-mediated stimulations[17], we, therefore, investigated it further. By analyzing galectin-1 concentrations in cell culture supernatants of *B. infantis*-stimulated T-cells, stimulation with *B. infantis* in comparison to *S. aureus* or non-specific stimulation (anti-CD3/CD28) resulted in a significant increase in galectin-1 in all age groups (Fig. 3c, Table S16).

To support the results of the RNA sequencing assays, we analyzed *B. infantis* stimulation of neonatal T-cells, which indeed led to an enhanced IL-10 accumulation in supernatants (Fig. 3d, Table S17). However, neutralization by specific antibodies for IL-10 - as verified by LEGENDplex assay (Fig. S7a; Table S57)—did not alter the parameters analyzed, such as proliferation, CD25 expressing cells, and differentiation of Th1 (IL-2, TNF-α, and IFN-γ) (Fig. 3e, f, Fig. S7b, Table S18 and 19, S58–61) or Th17 cells (Fig. S7c, Table S62 and 63).

Besides IL-35 (FDR < 0.05), one of our top hits being increasingly expressed was CTLA-4 (FDR < 0.0001). To decipher the role of it during *B. infantis*-specific stimulation, CTLA-4 blockage was performed using specific antibodies, and T-cell proliferation, expression of CD25 and Th1 cytokine producers (IL-2, TNF-α, and IFN-γ) were analyzed. Indeed, when checkpoint receptor CTLA-4 was blocked, enhancement of identified parameters was detected in all groups analyzed (Fig. 3e, f,

Fig. S7b). A profound effect of CTLA-4 blockage on *B. infantis*-specific stimulation was detectable for neonatal T-cell proliferation, which was tripled under CTLA-4 blockage, although only a small but nevertheless significant decrease in galectin-1 release was observed in neonatal and adult T-cells (Fig. S7d, Table S64). Comparing the effect of CTLA-4 blockage between T-cell responses to different bacteria showed that *S. aureus*-specific T-cell proliferation was much less dependent on CTLA-4 than *B. infantis*-specific response (Fig. 3g, Table S20).

As CTLA-4-dependency together with RNASeq data may point in the direction that Treg-like cells were generated after *B. infantis*-specific stimulation, we monitored the Treg master transcription factor FoxP3 in conjunction with transcription factors T-bet, GATA3, and RORγt as indicated. The data revealed that about 20% of FoxP3⁺ Tregs are formed in response to *B. infantis* irrespective of age, while in response to *S. aureus* this frequency drops significantly in adults in comparison to neonates (Fig. 3h, Tables S21 and 22). Furthermore, early in life, half of the induced FoxP3⁺ Tregs co-expressed FoxP3 with either GATA3, T-bet, or RORγt. However, exclusively in the case of *S. aureus* stimulation, FoxP3⁺ T-cells are constrained in a naïve T-cell pool of adults (Fig. 3h). Of note, many FoxP3⁺ T-cells appearing in *S. aureus*-induced T-cell pool co-express GATA3 implicating a high stability phenotype of these cells[35].

To determine whether the *B. infantis*-induced suppression of T-cells further impacts T-cell responses to other antigens, we exemplarily used neonatal and adult set-ups as described in Fig. 1. *B. infantis*-matured APCs were added 3 days after the beginning of the stimulation to *S. aureus* and *S. epidermidis* stimulated T-cells (Fig. 4a–e), which did not affect the activation, proliferation, or cytokine production profiles of these T-cells. However, T-cells that were stimulated vice versa initially with *B. infantis*-matured APCs followed by *S. aureus* (Fig. 4a–e) or *S. epidermidis* (Fig. 4f) matured APCs were no longer able to initiate a full-blown effector T-cell response. This was evidenced by measuring proliferation (Fig. 4a, b, f, Table S23–25), CD25 expression (Fig. 4c, Tables S26–28), and the frequency of Th1-like cytokine producers (Fig. 4d, Tables S29–31) of neonatal T-cells as well as of activated naïve and memory T-cells of adults. Concurrently, we observed that the initial contact with *B. infantis* antigens-presented on APCs led to a significant increase in released galectin-1 in naïve neonatal and adult T-cells as well as memory T-cells (Fig. 4e, Tables S31–34).

Since different probiotics such as *Bifidobacterium* or *Lactobacillus* have already been shown to shorten both the duration of respiratory symptoms and the days with fever caused by SARS-CoV-2, we investigated whether *B. infantis* can also attenuate the activity of T-cells of severely ill COVID-19 patients[36,37]. For this purpose, total CD4⁺ T-cells from severely ill SARS-CoV-2 infected patients were either stimulated with SARS-CoV-2 peptides- or *B. infantis*-matured monocytes overnight, respectively, alone or re-simulated halfway with the other antigen-matured monocytes as described in Fig. 4a–f for a total of

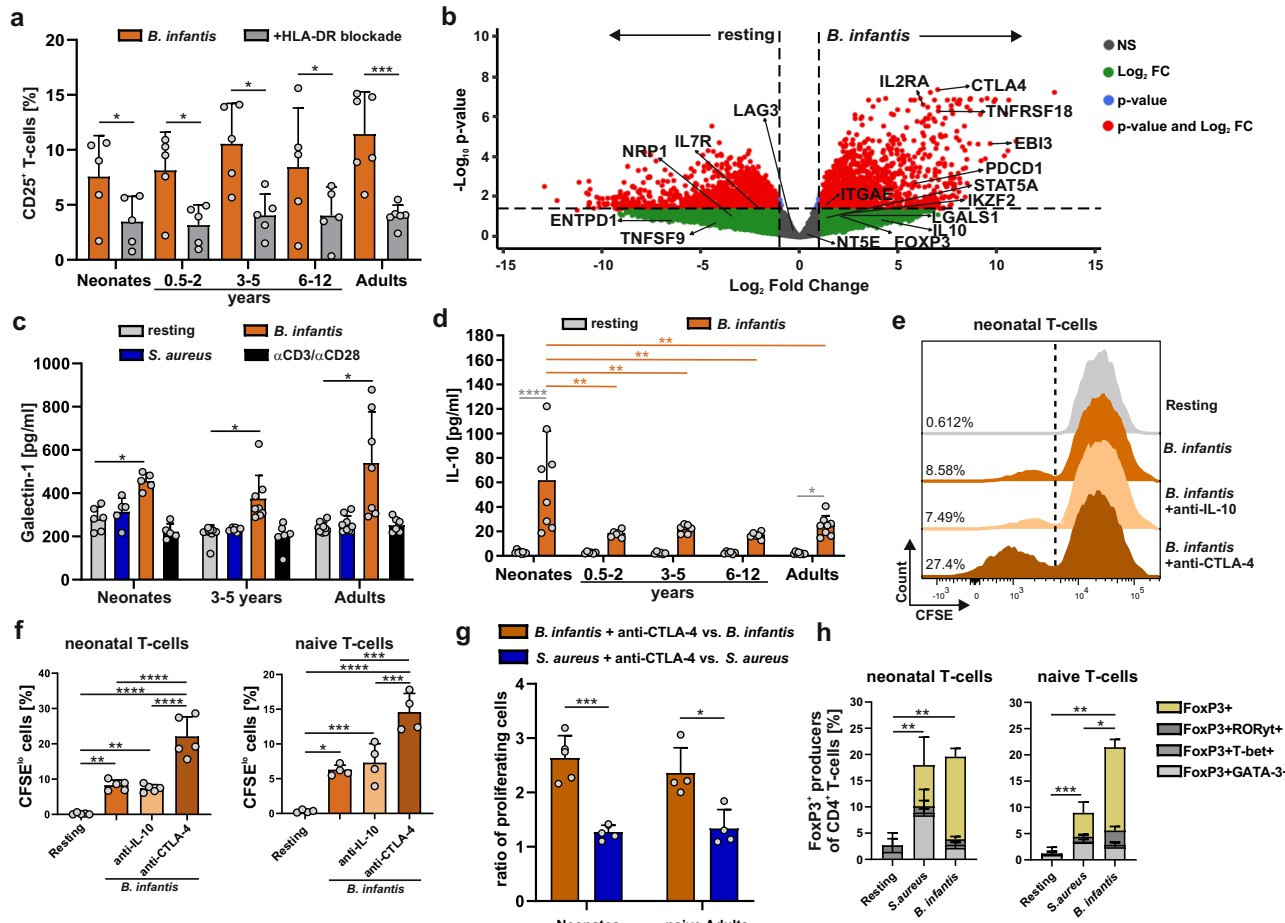

**Fig. 3 | *B. infantis*-specific human T-cell response. a** CD25 expression of naïve T-cells from neonates, infants, children, and adults in response to h.i. *B. infantis* matured monocytes for 3 days in the presence or absence of HLA-DR blocking antibody. **b** Volcano plot of RNASeq analysis showing differentially expressed RNA in neonatal T-cells in response to h.i. *B. infantis*-antigens. Three days after the beginning of the stimulation, T-cells were enriched using MACSQuantTyto, RNA was prepared, and RNASeq was performed as described in Materials and Methods. Red dots indicate genes that are significantly differentially regulated, while green dots indicate genes that are differentially regulated but not significantly. Data are representative of four donors. **c** Galectin-1 release from T-cells of neonates, children, and adults after 3 days of stimulation, as indicated, was measured by ELISA assay. **d** Determination of IL-10 cytokine release from CD4+CD45RA+CD31+ T-cells of neonates, infants, and children or adults either stimulated or resting for 3 days by LEGENDplex. **e** CFSE dilution profiles of neonatal CD4+CD45RA+CD31+ T-cells after 3 days of stimulation with monocytes matured with h.i. *B. infantis*-antigen in the presence or absence of anti-IL-10 antibody or anti- CTLA-4 antibody. **f** Bar graphs showing the frequency of CFSE^lo T-cells from neonates (left) and of naïve T-cells from adults (right) as determined by flow cytometry. **g** Bar graph showing the ratio of proliferating (CFSE^lo) T-cells in response to h.i. *B. infantis*-antigens (orange) or h.i. *S. aureus*-antigens (blue) in the presence of anti-CTLA-4 antibody and in response to *B. infantis*-antigens or *S. aureus*-antigens alone. **h** Frequencies of T-cells expressing intracellular FoxP3 (yellow), FoxP3/RORγt (dark gray), FoxP3/T-bet (middle gray), and FoxP3/GATA-3 (light gray) were measured by flow cytometry after 3 days of stimulation, analyzed by Boolean gating and presented as fractions of cells expressing transcription factors in a stacked bar chart. Error bars in the figures (**a, c, d, f, g**) denote mean + SD, $n \geq 5$ donors from at least 3 independent experiments in each age group, *p*-values in (**a, c, d, f, g**) are calculated by one-way ANOVA followed by Holm−Sidak post hoc test, *$p < 0.05$, **$p < 0.01$, ***$p < 0.001$, ****$p < 0.0001$. Detailed statistical data are provided in Supplementary Tables 14−22. Source data are provided as a Source Data file.

6 days and analyzed by flow cytometry (Fig. 4g, h, Table S35−38). In the presence of SARS-CoV-2 peptides, T-cells from COVID-19 patients responded with markedly enhanced proliferation ($p < 0.0001$) (Fig. 4g, left), associated with dramatically increased Th1-cytokine producers (Fig. 4g, right). In contrast, although *B. infantis*-enhanced T-cell responses were observed, these were strikingly decreased in comparison to those of SARS-CoV-2 peptides, except in terms of IL-10 secretion (Fig. 4g, h). In a similar manner to our previous infection models (Fig. 4a−f), the addition of SARS-CoV-2 peptides matured monocytes to *B. infantis* stimulated T-cells no longer augmented a full-blown effector T-cell response. Intriguingly, upon re-exposure to *B. infantis*-matured monocytes, T-cells stimulated with SARS-CoV-2 peptides also showed suppressed proliferation and a dramatic reduction in Th1-cytokine producers by almost 50% ($p = 0.023$), associated with enhanced IL-10 and galectin-1 secretion (Fig. 4g, h). Having shown

in Fig. 3g and e that the inhibitory function of T-cells exposed to *B. infantis* is CTLA-4 dependent, we analyzed this dependence in the context of SARS-CoV-2-specific T-cell responses of severe COVID-19 patients from ICU (Fig. 4i). Indeed, blockage of CTLA-4 during inhibition of *S. aureus*-stimulated T-cells using specific antibodies revealed a complete reversal, indicating a requirement for cell−cell contact for the inhibitory function of *B. infantis*-primed T-cells. As heating of *B. infantis* may have removed some of the volatile metabolites produced by the microorganism, we analyzed live *B. infantis* for mediation of T-cells with suppressive capacity (Fig. 5a, b, Tables S39 and 40). For this purpose, we matured neonatal and adult monocytes with washed live *B. infantis*. These *B. infantis*-matured APCs were added 3 days after the beginning of the stimulation to *S. aureus*-stimulated T-cells, as described in Fig. 4. Compared to *S. aureus* alone, initial contact with live *B. infantis* led to a dramatic reduction in proliferation of about 60%

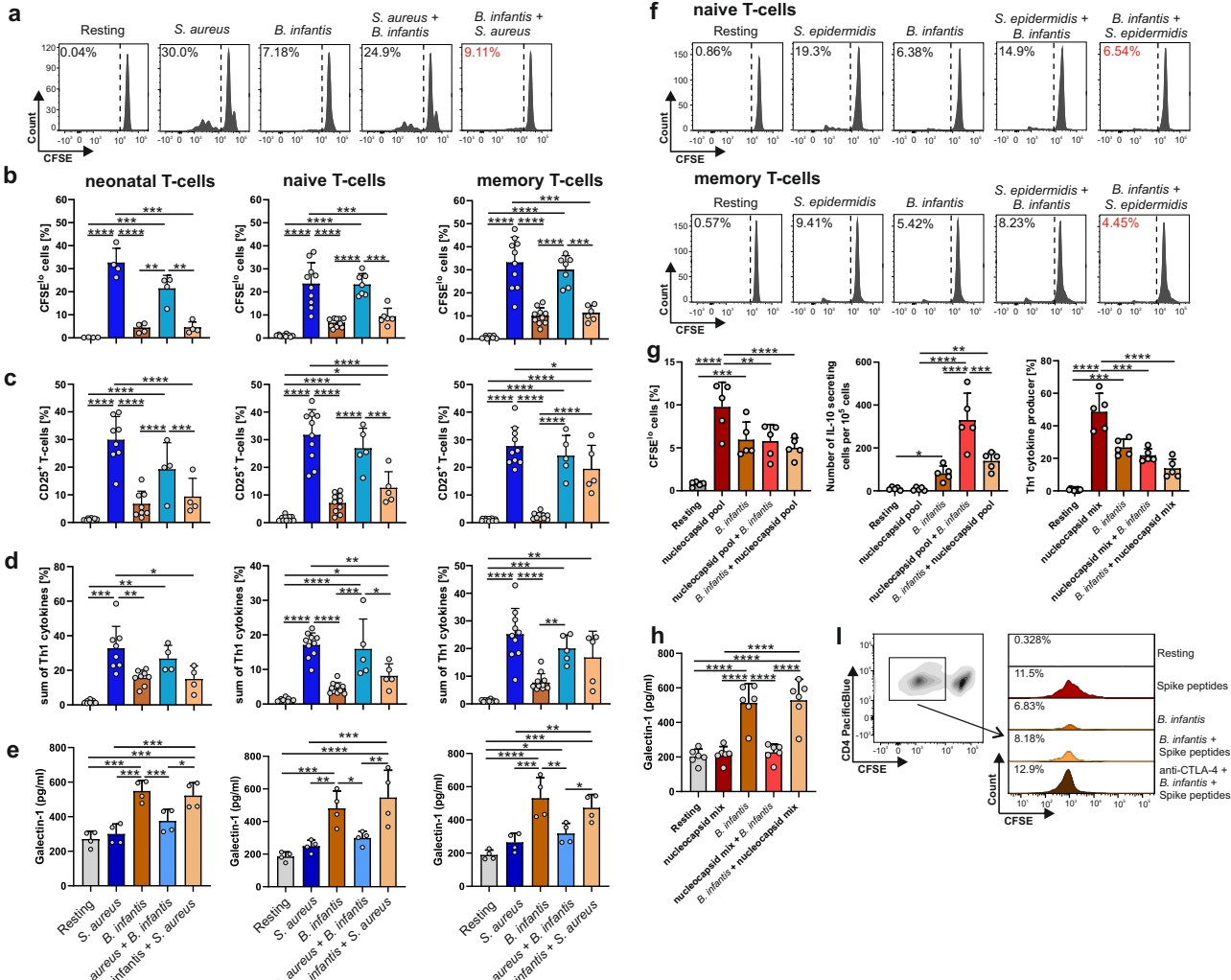

**Fig. 4 | *B. infantis's* role in mediating suppression. a** Purified CD4⁺CD45RA⁺CD31⁺ T-cells of neonates were labeled with CSFE and co-cultured for 3 days with *S. aureus*- or *B. infantis*-matured monocytes. T-cells were further cultured for 3 days with monocytes that had been equally loaded or cross-loaded. CFSE dilution profiles on day 3 after stimulation. Data are representative of four donors out of two experiments. **b–e** Neonatal and adult CD4⁺CD45RA⁺CD31⁺ T-cells, as well as adult CD4⁺CD45RO⁺ T-cells were stimulated as in (**a**). Frequency of proliferating (CFSE^lo) T-cells (**b**), of T-cells expressing CD25 (**c**), or the sum of T-cells expressing intracellular IL-2, TNF-α, and IFN-γ (**d**) was determined by flow cytometry and concentrations of galectin-1 in supernatants (**e**) using ELISA. Data are representative of four donors out of two experiments in each age group. (**f**) Purified naïve or memory adult T-cells were labeled with CSFE and co-cultured with monocytes matured with *S. epidermidis* or *B. infantis* or both, as indicated. CFSE dilution profiles of a representative neonatal donor on day 3 after stimulation. Data are representative of three donors out of two experiments. **g** CD4⁺ T-cells from five patients with severe

COVID-19 were co-cultured in three independent experiments with monocytes matured with SARS-CoV-2 peptide pool of nucleocapsid protein or *B. infantis* or both. Frequency of CFSE^lo T-cells (*left*), quantitative IL-10 production by EliSpot analysis (*middle*), or the sum of T-cells expressing intracellular IL-2, TNF-α, and IFN-γ (*right*) were determined by flow cytometry and galectin-1 concentrations (**h**) in supernatants by ELISA. Error bars in figures denote (**b**, **c–e**, **h**) mean + SD, *n* ≥ 4 donors from at least 2 independent experiments in each age group, *p*-values in (**b**, **c–e**, **h**) are calculated by one-way ANOVA corrected by Holm−Sidak post hoc test, *$p < 0.05$, **$p < 0.01$, ***$p < 0.001$, ****$p < 0.0001$. Detailed statistical data are provided in Supplementary Tables 23–38. **i** CFSE dilution profiles of CD4⁺ T-cells from a patient with severe COVID-19 after stimulation with monocytes matured with SARS-CoV-2 peptide pool derived from spike protein or *B. infantis* or both in the presence or absence of anti-CTLA-4 antibody. Data are representative of four donors out of at least two experiments. Source data are provided as a Source Data file.

in neonatal T-cells. Indeed, neonatal and adult naïve T-cells responded similarly to h.i. or live *B. infantis* matured monocytes (Fig. 3) and mediated suppression of *S. aureus*-specific T-cell responses. This suggests that h.i. did not destroy the potency of *B. infantis* to mediate T-cell differentiation with suppressive potential.

As we have shown that blockage of CTLA-4 at least partially restores the suppressive function of *B. infantis*, implying that cell–cell contact is likely to be required, we next monitored cell-cell contact more closely in terms of proliferation of target T-cells (Fig. 5c, Tables S41 and 42). Therefore, we labeled neonatal and adult T-cells with CFSE and stimulated the T-cells with monocytes matured with

either *S. aureus* or *B. infantis* in transwell chambers where soluble factors could be exchanged. Notably, soluble factors released by proliferating *S. aureus*-specific T-cells could not enhance proliferation against *B. infantis*. In fact, no inhibition of the *S. aureus* response occurred upon separation, reinforcing our findings that cell–cell contact is obligatory for the fully inhibitory function of *B. infantis*-differentiated T-cells. To confirm our finding that neonatal T-cells exposed to *B. infantis* are Treg-like, we subjected them to a standard suppression assay (Fig. 5d). Indeed, using a suppression assay, we were able to confirm that *B. infantis* differentiated neonatal T-cells (Spearman's Rho = −0.895, *p* < 0.001) are able to inhibit T-cells that are non-

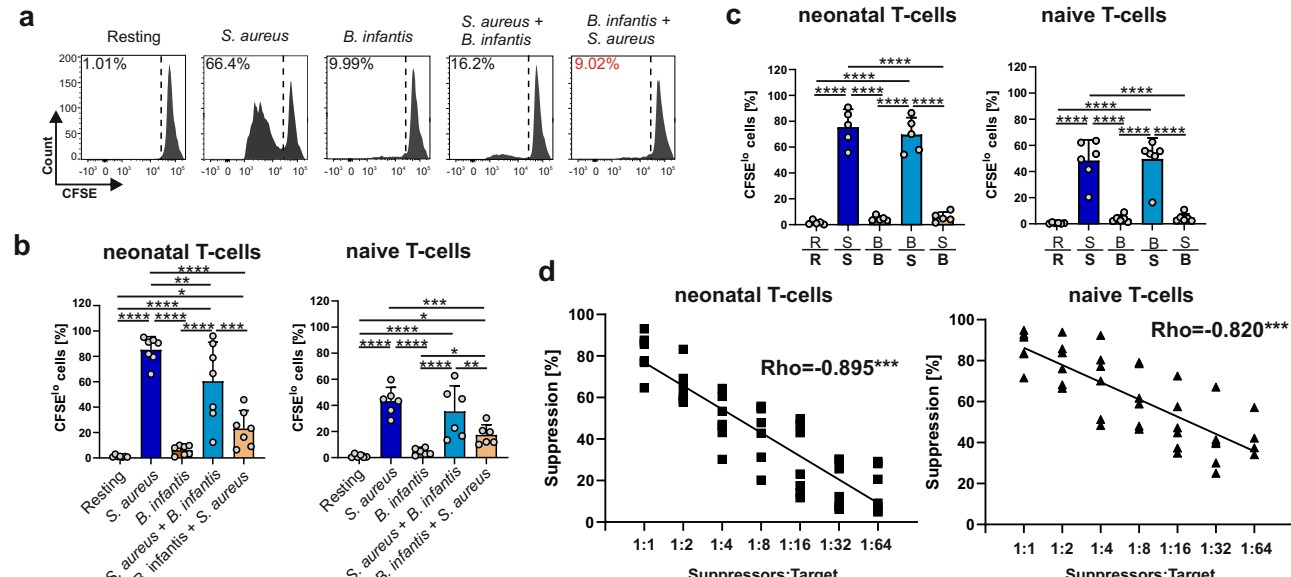

**Fig. 5 | Regulatory role of *B. infantis* specific T-cells. a** Purified neonatal CD4+CD45RA+CD31+ T-cells were labeled with CSFE and co-cultured for 3 days with monocytes matured with h.i. *S. aureus* or living *B. infantis* as indicated. T-cells were further cultured for 3 days with monocytes that had been equally loaded or cross-loaded. CFSE dilution profiles on day 3 after stimulation. Data are representative of 7 donors from 5 independent experiments. **b** Neonatal and adult CD4+CD45RA+CD31+ T-cells were stimulated with h.i. *S. aureus* or living *B. infantis* for 3 days as in (**a**). The frequency of proliferating (CFSElo) T-cells was determined by flow cytometry. **c** Statistical analysis of Transwell assays with purified naïve (CD4+CD45RA+CD31+) T-cells of neonates and adults. Thereby, the content of the upper chamber is indicated above the line, and the content of the lower chamber is indicated below the line. T-cells in the lower chamber were labeled with CFSE and co-cultured for 6 days, followed by analysis by flow cytometry (R—Resting, S—*S. aureus*, B—*B. infantis*). Error bars in figures (**b**, **c**) denote mean + SD, $n \geq 5$ donors from at least 3 independent experiments in each age group, *p*-values in (**b**, **c**) are calculated by one-way ANOVA followed by Holm−Sidak post hoc test, *$p < 0.05$,

**$p < 0.01$, ***$p < 0.001$, ****$p < 0.0001$. Detailed statistical data are provided in Supplementary Tables 39–42. **d** Neonatal and adult CD4+CD45RA+CD31+ T-cells were co-cultured for 3 days with monocytes matured with h.i. *B. infantis*, and assayed for suppressive activity using CFSE-labeled resting target T-cells and different ratios of T-cells stimulated prior assay onset for 3 days with *B. infantis*-matured monocytes. The mix of cells was then stimulated with anti-CD3 antibody in the presence of dendritic cells for 4 days, and proliferating cells were determined based on a decrease in CFSE expression. Percent suppression was calculated based on CFSE dilution of target T-cells in suppression culture. Cumulative results are shown, and each dot represents one donor. Correlation analyses were performed using Spearman´s Rho, with *** representing $p < 0.001$. The lines show a linear approximation of the logarithmically transformed ratio of T suppressor cells to target cells. The data shown are representative of at least five experiments performed with T-cells from different donors. Source data are provided as a Source Data file.

specifically stimulated with anti-CD3/CD28, to a similar extent as published studies using Treg cells[38].

Thus, using human infection models (Figs. 3 and 4), our data clearly demonstrate that *B. infantis*-stimulated T-cells can actively dampen an exuberant immune response, most likely by inducing Treg formation.

## Discussion

Here, we report that early in life, a strong pro-inflammatory anti-bacterial T-cell response can be initiated against *S. aureus* and *S. epidermidis* antigens, whereas the response observed against *B. infantis* was anti-inflammatory. These different responses reflect well the immunological requirements in the first years of life, when immuno-logical memory is not yet sufficiently acquired: immediate defense mechanisms are required to protect the organism, while concomitant inflammation must be kept low to prevent damage to developing tissues.

In our study, we found that pro-inflammatory bacteria can elicit strong antigen-specific T-cell responses in neonates, who exhibit a greater response than children and adults, potentially due to a lowered activation threshold[6]. T-cells may take over memory-like features at birth to compensate for the lack of memory T-cell respones[39], similar to virtual memory T-cells[40]. At least in mice, these virtual memory T-cells arise from pre-selected thymic emigrants. Another explanation for higher frequencies is the broader T-cell repertoire at birth, as hardly any T-cells have yet switched to the memory compartment. T-cells with unique propensities from the neonatal compartment have

been identified in the adult memory compartment[7]. In line with this assumption is that frequencies of naïve precursors determine the magnitude of the T-cell response[41]. At birth, the monitored enhanced frequencies and reactivity of bacterial-specific T-cells could also be explained by being exposed to elevated concentrations of pro-inflammatory IL-6 and IL-1β in the blood during the delivery time frame. This milieu is able to put T-cells in an alerting Th pre-cursor-like state[42]. However, *B. infantis* elicits immunosuppressive T-cell responses in neonates. The generation of pro-inflammatory versus regulatory T-cells may be influenced by the strength and type of inflammatory context, as well as a certain localization[43]. Another explanation would be that it is potentially due to the generation of selected precursors within the thymus[40]. One study even suggests that the origin of neonatal thymic emigrants from the fetal liver, not as later in life from bone marrow, gives the bias for Treg generation in the periphery[44]. Some bacteria may have induced TCR repertoire skewing in a symbiotic context due to their close evolutionary rela-tionship with humans, evidenced by tailored nutrients for *B. infantis* found in human milk[13,14]. TCR repertoire skewing suggested to ori-ginate in the thymus, has not only been observed in localized iTregs in the gut, which require commensal bacteria for their generation[43], but also in Enterobacteriaceae-specific Th cells[45]. According to recent results, the different Tconv and Treg-biased responses might be associated with the selection/skewing of different regions of the TCR, CDR3β hydrophobicity promotes Treg cells, while the *TRBV* gene shapes the TCR's general activatability[46]. In fact, these scenarios that could lead to different T-cell responses to different bacteria -

selection, bias, and diverse biological contexts—are likely to be intertwined.

Here we show a high potential of naïve T-cells to develop a regulatory phenotype when exposed to *B. infantis* presented by APCs (Fig. 3b). These T-cells demonstrating suppressive capacity in antigen-specific and non-specific suppression models (Fig. 4) are characterized by upregulation of genes encoding regulatory proteins such as the major Treg transcription factor FoxP3, signatory molecules such as GITR and Helios and secretion of immunosuppressive cytokines (IL-10, IL-35) and metabolites (galectin-1)[34]. A direct cell-cell contact dependency by CTLA-4 was shown, which is known to enhance the Treg-mediated suppression of effector T-cell functions[47,48] and their homeostasis (Fig. 3b, Fig. S5)[49]. Consistent with other studies showing strong upregulation of galectin-1 mRNA in tissue samples in response to *B. infantis*[17], we observed increased levels of secreted galectin-1 protein in T-cells recognizing *B. infantis* (Fig. 3c, Fig. S7c, Fig. 4h) that likely acts to inhibit effector T-cell activation[50] and to induce Treg cells[42,51–53]. Enhanced and stable FoxP3+ Tregs in response to *B. infantis* also suggest the generation of a T-cell subset, which may play an important role in suppressing autoimmune pathologies (Fig. 3, Fig. 4). Of note, other strains of *Bifidobacterium* probably mediate different, but also inhibitory responses, as administration of *Bifidobacterium lactis* led to an increase in monocyte and killer cell activity in elderly subjects[54].

Our results further show an IL-17A signature cytokine expression at birth in antigen-specific T-cell responses to *S. epidermidis* and *S. aureus* that is corroborated by the master transcription factor for Th17 cells, RORγt (Fig. 2h). Despite the induction of an IL-17A program, we show a high potential of naïve T-cells to express FoxP3, which is the master transcription factor of Tregs. This suggests at the cellular level that during the time window in early life when Tregs are preferentially induced, microbe-specific T-cell responses serve as triggers for Treg differentiation[51]. As the inhibition of TLRs 1–4, 7, 8 (T-cells do not express TLRs 3, 8, 9) and complement receptors did not alter the response (Fig. S3), the sensitivity of T-cells early in life to become Th17 cells or Tregs is likely intrinsic and/or due to different origin in cell lineages at birth[44,55]. Whether these cells are also the FoxP3+ Tregs that *B. infantis* mediates later in life (Fig. 3f) remains elusive. Even though, at a lower level, *S. aureus*-mediated differentiation of adult CD4+FoxP3+ T-cells includes about 50% GATA3 co-expressing cells, which especially renders these Tregs stable[55]. Indeed, if different differentiation requirements or origins are considered early in life, it is likely these generated cells may have distinct qualities, e.g., not all Th17 cells are functionally similar and depend on differentiation peculiarities[56]. In line with this, as shown in mice, Tregs against *S. epidermidis* that infiltrate the skin can only be generated in a crucial time window early in life[57]. Thus, the timing of first contact with a certain pathogen or antigen matters; it profoundly impacts the immune response after re-encounter in adulthood[51,58].

From 6 years on, naïve T-cell responses remained similar against *S. aureus* and *S. epidermidis* in showing an increased Th1-like response. As IL-17 is considered to be a signatory cytokine for neonates, our data show that, indeed, its preference is antigen-specific and continues while infancy proceeds. Likely, losing the bias for IL-17 responses unequivocally leads to the default Th1 pathway. As T-bet-expressing T-cells are as frequent in the neonatal T-cell pool as in the adult one (Fig. 2c, left), whereas IFNγ secretion is lower in neonatal and infant T-cells upon antibacterial stimulation (Fig. 2b, right, Fig. 2d), the availability of the IFN-γ promoter may be limited. This is consistent with the notion that the IFN-γ promoter has been described to be hypermethylated in human neonatal T-cells and that T-cell non-specific and fungal-specific stimulation showed reduced IFN-γ production in children up to 12 years of age[7,59]. As IFN-γ suppresses IL-17-responses,

reduced availability of the IFN-γ machinery in antibacterial-specific responses likely contributes to the bias towards IL-17 expression[60].

The preference for IL-17 responses is lost earlier in *S. epidermidis* reactive T-cells than in *S. aureus*-specific ones, and still with both of them years earlier than reported of *Candida albicans*-specific T-cell responses[7]. Furthermore, we observed a weak IL-17 expression in response to *B. infantis*. These data are in line with recent findings that *B. infantis* silences Th2 and Th17 responses in breastfed infants[17]. Of note, we cannot rule out an enhanced plasticity of the Th17 cells to become Th1-like cells with increasing age. Another explanation would be that differences are due to the migration of *S. epidermidis*-specific Th17 cells into tissues[57]. With the specific focus here on naïve T-cell differentiation, a prior unrecognized transfer into the memory pool is unlikely as germinal centers are not fully developed before the age of two[61]. Interestingly, even though Th17 cells dominate the anti-*S. aureus* response up to 5 years of age, only 50% of Th1 cells are cytokine co-producers/multifunctional T-cells that tend to enter the memory pool[27].

With regard to the importance of *Bifidobacterium* in SARS-CoV-2 infections, we could show that *Bifidobacterium longum infantis* is able to modulate the activity of SARS-CoV-2 nucleocapsid-specific CD4+ T-cells (Fig. 4). These findings might explain previous observations that the abundance of *Bifidobacterium* is likely negatively associated with severity and duration of SARS-CoV-2 infection[62,36]. However, CD8 T-cells, the major players in viral infections, were not included in our human model. Nevertheless, it can be speculated that at least a mediated reduction in Th1-cell help may ultimately lead to altered CD8 activity. *Bifidobacterium* may therefore reduce the severity of COVID-19[63,64]. Whether this is a general mechanism for *Bifidobacterium*-mediated protection against viral infections, as has been suggested for the influenza virus, needs to be pursued further[65]. However, the evidence provided here that *B. infantis*-stimulated T-cells can act broadly to suppress adverse immune responses, such as Th-cell responses against *S. aureus*, *S. epidermidis* and SARS-CoV-2, but most importantly, in a standard suppression assay (Fig. 5d), supports this notion. Just as known for Treg cells, the *B. infantis*-specific T-cells described here are antigen-specifically induced and exert their suppressive function in a specific and/or bystander manner[66].

Taken together, our data show that *S. aureus*, *S. epidermidis*, as well as *B. infantis* are recognized by T-cells in an antigen-specific manner and that activation clusters form in a similar way. The features of *B. infantis*-specific T-cells, including the expression of Treg-associated molecules, metabolic profile, and suppressive function, suggest the induction of Tregs or the activation of inhibitory mechanisms. This is supported by the fact that suppression of effector cells can be partially reversed through CTLA-4 blockage, one core of mechanisms contributing to Treg-mediated suppression[67]. In addition, *B. infantis* is powerful to induce IL-10 production in activated T-cells. It is reasonable to speculate that this strong IL-10 production is responsible for anti-inflammatory effects not examined here, as has been elegantly demonstrated in a mouse model for the mitigation of gut immunopathologies[20]. Thus, *B. infantis* could be a crucial factor in counteracting immune pathologies during the window of opportunity of the first months of life in inducing tolerance and could be exploited at any age as a tool to alleviate adverse overreactions of unwanted immune responses.

## Methods

### Ethics statement

The study was reviewed and approved by the Clinical Research Ethics Committee of the University of Magdeburg (certificates 06/11, 79/07, 26/12, and 159/18), and all healthy donors, parents, or relatives of severe COVID-19 patients gave written informed consent in accordance with the Declaration of Helsinki.

## Samples

Umbilical CB samples were collected from the umbilical veins of 23 term newborns (9 females, 13 males, 1 n/a) immediately after birth from the St. Marienstift Hospital, Magdeburg, Germany (Table S1). Gestational age ranged from 37 to 41 weeks (median gestational age: 39 weeks). PB and adenoids were obtained from children (aged 0.5–12 years) with adenoid hypertrophy by surgical excision and were provided by the Department of Otorhinolaryngology, University Hospital of Magdeburg (Table S2). Children with immune or genetic disorders were excluded. All children were clinically free of infection at the time of surgery, were not taking any medication, and had no chronic disease. PB mononuclear cells (PBMCs) were obtained from leukocyte reduction filters (Sepacell RZ-2000; Asahi Kasei Medical) of 23 healthy adult donors (7 females, 17 males, age range 20–59 years, median age: 44 years) provided by the Institute of Transfusion Medicine and Immunohematology, University Hospital Magdeburg, Germany (Table S3). In addition, we enrolled 5 acute severe COVID-19 patients (3 women, and 2 men, age range 45–72 years, median age: 69 years) from the intensive care unit (ICU) of the University Hospital Magdeburg (Table S3). Blood samples were taken in March and April 2021. COVID-19 patients were tested positive for SARS-CoV-2 RNA and/or anti-SARS-CoV-2 spike abs within 3 days prior to blood collection.

## Cell purification and cell culture

Mononuclear cells were obtained from PB of healthy donors, surgically excised adenoids of infants suffering from non-inflammatory hypertrophy, or CB by centrifugation of Ficoll–Hypaque gradient. CD14$^+$ monocytes isolated from CB, blood of infants, and healthy donors using CD14-MicroBeads (Miltenyi Biotec) and autoMACS-Pro isolation (Miltenyi Biotec) were matured with heat-inactivated (h.i.) *Staphylococcus aureus ssp. aureus Rosenbach* (ATCC 25923, no toxin-producing strain), h.i. *Staphylococcus epidermidis (Winslow and Winslow) Evans* (ATCC 12228), h.i. or alive *Bifidobacterium longum ssp. infantis* (ATCC 15697) or SARS-CoV-2-peptide pool PepTivator SARS-CoV-2 Prot_S derived from spike protein or PepTivator SARS-CoV-2 Prot_N derived from N protein (both pools contain lyophilized peptides consisting mainly of 15-mer sequences with 11 amino acid overlap and are used at a concentration of 18.75 ng ml$^{-1}$, Miltenyi Biotec) overnight at 37 °C in RPMI 1640 (PAN Biotech) containing 10% fetal bovine serum (Gibco/Life Technologies GmbH); 10 μg ml$^{-1}$ streptomycin; and 10 U ml$^{-1}$ penicillin (Life Technologies GmbH). Microbes were washed three times with PBS and centrifuged at 4000xg in between and then killed by heating at 65 °C for 1 h (h.i.) according to standard methods, followed by three freeze-thaw cycles. Protein concentration was determined by the bicinchoninic acid assay (Bio-Rad) according to the manufacturer's instructions. The concentration of h.i. bacteria used in co-culture experiments were determined after titration in T-cell proliferation assays using CFSE-labeled T-cells, as described below. 5 μg ml$^{-1}$ of h.i. bacteria *S. aureus*, 5 μg ml$^{-1}$ *S. epidermidis* or 10 μg ml$^{-1}$ *B. infantis* gave max. proliferative response and viability of T-cells when applied to $2.5 \times 10^5$ monocytes per ml for 16 h (Fig. S2B)[12]. H.i. bacteria were tested for endotoxin concentrations using Pierce LAL Chromogenic Endotoxin Quantitation Kit (Thermo Scientific) and used with less than 0.036 EU ml$^{-1}$ in bacteria-monocytes co-cultures. Monocytes were then washed twice prior to co-culturing with T-cells in a 1:2 ($2.5 \times 10^5$ monocytes ml$^{-1}$ and $5 \times 10^5$ T-cells ml$^{-1}$). CD4$^+$CD45RA$^+$ T-cells or recent thymic emigrants (CD4$^+$CD45RA$^+$CD31$^+$) were enriched to high purity (>99.6%) by magnetic beads separation with autoMACS-Pro using human naive CD4$^+$ T-cell Isolation Kit or human CD4$^+$ Recent Thymic Emigrant Isolation Kit (Miltenyi Biotec), respectively.

A total of $5 \times 10^5$ cells ml$^{-1}$ enriched CD4$^+$CD45RA$^+$ T-cells or recent thymic emigrants (CD4$^+$CD45RA$^+$CD31$^+$) were stimulated with the bacteria-matured CD14$^+$CD16$^+$ monocytes at a ratio 2:1. For blockage of HLA-DR, anti-HLA-DR (10 μg ml$^{-1}$, L249, homemade, controlled by

Western blotting and competitive FACS analysis) blocking antibody was used. IL-10 was neutralized by using an anti-IL-10 mab (10 μg ml$^{-1}$, JES3-19F1, Biolegend), and blocking was controlled by LegendPlex (Fig. S7A). For TLR inhibitor experiments, MyD88 inhibitor Pepinh-MYD (50 μM, InvivoGen) was used. For blockage of CR3α, anti-CD11b (10 μg ml$^{-1}$, ICRF44, Biolegend) and anti-CD18 (10 μg ml$^{-1}$, TS1/18, Biolegend) blocking antibody were used. TGFß blockage was performed by anti-TGFß (10 μg ml$^{-1}$, 19D8, Biolegend) and blockage of CTLA-4 by anti-CTLA-4 (10 μg ml$^{-1}$, BNI3, BD Pharmingen). For positive control, T-cells were stimulated routinely with microbeads coated with anti-CD3 (1 μg ml$^{-1}$; UCHT1) and anti-CD28 (2 μg ml$^{-1}$; CD28.2) (both Biolegend) at a cell-to-bead ratio of 2:1.

## Flow cytometric analysis

Cytometric analyses were performed using a FACS Canto II (Becton Dickinson) or FACS Fortessa X-20 (Becton Dickinson) and FACS Diva software version 9.0.1 (BD Biosciences). To determine cell proliferation, T-cells were labeled with CFSE (Molecular Probes/Life Technologies GmbH) according to the standard protocol. Prior to intracellular cytokine analysis, enriched CD4$^+$CD45RA$^+$ T-cells or recent thymic emigrant (CD4$^+$CD45RA$^+$CD31$^+$) T-cells ($5 \times 10^5$ cells ml$^{-1}$) were stained after restimulation with 10 ng ml$^{-1}$ PMA (Sigma-Adrich) and 1 μg ml$^{-1}$ ionomycin (Cell Signaling) for 3 h plus adding 5 μg ml$^{-1}$ brefeldin A (Cell Signaling) for the last 2 h. Cells were fixed with 2% paraformaldehyde (Morphisto GmbH) in PBS for 20 min and permeabilized with 0.5% saponin (Sigma-Aldrich) in PBS/BSA, and incubated with the following antibodies: anti-IL-17A (eBio64DEC17, eBioscience), anti-IFN-γ (4 S.B3; 1:1000; BD Biosciences), anti-IL-10 (JES3-D7; 1:100; BD Biosciences), anti-CD4 (RPA-T4; 1:100), anti-CD3 (SK7; 1:100); anti-CD25 (BC96; 1:100), anti-CD45 (HI30; 1:100), anti-IL-2 (MQ1-17H12; 1:100), anti-TNF-α (Mab11; 1:100) (all Biolegend). For the detection of (pro-)apoptotic T-cells, cells were stained with propidium iodide (PI) and annexin V in binding buffer (10 mM Hepes, pH 7.4, 140 mM NaCl, 2.5 mM CaCl$_2$) or in blocking buffer (10 mM Hepes, pH 7.4, 140 mM NaCl, 2 mM EGTA) as a control. AnnexinV and PI-negative cells were considered viable.

To analyze activation-induced surface molecules by flow cytometry, cells were harvested and stained with anti-CD4 (RPA-T4; 1:100), anti-CD3 (SK7; 1:1000); anti-CD45 (HI30; 1:100), anti-CD25 (M-A251; 1:100) and anti-CD69 (FN50; 1:100) (all Biolegend) in PBS/BSA. To measure transcription factors by flow cytometry, cells were fixed with 4% formaldehyde (Merck) in PBS for 10 min at 37 °C followed by permeabilization in ice-cold 90% methanol (Carl Roth) for 30 min and then stained with antibodies for specific transcription factors (anti-RORγt; 1:100 (BD Bioscience), anti-T-bet, anti-GATA-3 and anti-FoxP3 (all 1:100; all Biolegend)).

For data analysis, FlowJo software (version 10.8.1, Treestar) was used. Dead cells were excluded by forward and sideward scatter gating (Fig. S2A). The percentage of cytokine-producing CD4$^+$ T-cells was evaluated in the live CD3$^+$CD45$^+$ gates. Polyfunctionality was assessed using expression of the three cytokines IFN-γ, IL-2, and TNF-α. The combination of all Boolean gates was generated in FlowJo, analyzed, and graphically represented using Prism 9 (GraphPad Software Inc.).

## Cluster analysis

To detect the numbers of clusters/aggregates formed by cells after stimulation as described above of CD4$^+$CD45RA$^+$CD31$^+$ T-cells from neonates and adults, T-cells were labeled with CFSE (Molecular Probes/Life Technologies GmbH) according to the standard protocol prior to stimulation as described above. CFSE-labeled cell clusters have been counted by ImmunoSpot S6 ULTIMATE UV Image Analyzer.

## Analysis of cytokines in T-cell culture supernatant

Levels of IFN-γ, TNF-α, IL-2, IL-6, and IL-10, in human T-cell assay supernatants were measured using LEGENDplex human Th Cytokine

Panel (Biolegend) according to the manufacturer's instructions. Analyses were performed using FACSCanto (Becton Dickinson) and analyzed with LEGENDplex Data Analysis Software Suite (Qognit).

## Analysis of galectin-1 in T-cell culture supernatant

Cell culture concentrations of galectin-1 protein were determined by ELISA using commercially available kits (R&D Systems, Minneapolis, USA) according to the manufacturer's instructions. Samples were diluted 1:100 in assay buffer. The samples, along with the standards provided for each kit, were analyzed using an ELISA plate reader (MULTISKAN FC, Thermo Scientific).

## EliSpot assays

The commercially available Human IFN-γ/IL-10 Double-Color Enzymatic ELISPOT Assay (CTL-Europe GmbH) was used to measure IL-10 secretion of single cells. T-cells were co-cultured with antigen-pulsed monocytes for 3 days at 37 °C/5% $CO_2$. EliSpot plates were coated with human IL-10 Capture Antibodies. T-cells were transferred to EliSpot plates and incubated for 24 h at 37 °C/5% $CO_2$. For detection, biotinylated anti-human IL-10 Detection Antibodies were used. Spots were developed according to the manufacturer's instructions and counted with ImmunoSpot S6 ULTIMATE UV Image Analyzer.

## Fluorescence-activated cell sorting

To separate the matured monocytes from T-cells after 3 days of co-culture, cells were harvested and stained with anti-CD4 (REA613; 1:100), anti-CD3 (REA623; 1:100), and anti-CD14 (REA599; 1:100) (all Miltenyi Biotec) in MACSQuant Tyto Running Buffer. Cell sorting was conducted using a MACSQuant Tyto Cell Sorter (Miltenyi Biotec).

## Next generation sequencing

RNA extraction of sorted T-cells was performed with a NucleoSpin RNA isolation kit (Macherey-Nagel) according to the manufacturer's instructions. RNA integrity of RNA extracted from neonatal and adult T-cells activated with monocyte matured with h.i. *B. infantis* was controlled on a 4200 TapeStation System (Agilent Technologies). The RNA library was prepared using the Collibri 3′ mRNA Library Prep Kit for Illumina Systems (Invitrogen) according to the manufacturer's instructions. Sequencing was performed with the NextSeq 500/550 High Output Kit v2.5 (75 Cycles) on a NextSeq 550 Sequencer (Illumina) with single read 75 bp and 6 bp index read.

Each FASTQ file gets a quality report generated by the FASTQC tool. Before alignment to the reference genome, each sequence in the raw FASTQ files was trimmed on base call quality and sequencing adapter contamination using Trim Galore! wrapper tool. Reads shorter than 20 bp were removed from the FASTQ file. Trimmed reads were aligned to the reference genome using open source short read aligner STAR (https://code.google.com/p/rna-star/) with settings according to the log file. Feature counts were determined using the R package Rsubread. Only genes showing counts greater than 5 at least two times across all samples were considered for further analysis (data cleansing). Gene annotation was done by R package bioMaRt. Before starting the statistical analysis steps, expression data was log2 transform and TMM normalized (edgeR). Differential gene expression was calculated by R package edgeR. Functional analysis was performed by R package clusterProfiler.

## Suppression assay

Neonatal and adult CD4$^+$CD45RA$^+$ CD31$^+$ T-cells were sorted as described above and co-cultured for 3 days with monocytes matured with h.i. *B. infantis* and used as suppressor cells (Treg) in the further experiment. Further naive neonatal and adult T-cells were labeled with CFSE and used as target cells. The CFSE-labeled resting T-cells and the unlabeled Treg cells were mixed in different Treg-to-target ratios. Monocytes loaded with soluble anti-CD3 antibody (UCHT1,

50 ng/ml) for 1 h, 37 °C, $CO_2$ before T-cells were added. Cells were harvested on day 4 post activation and analyzed by FACS (Fortessa X-20, BD Biosciences) and for T-cell proliferation determined as CFSE-labeled target T-cells. The percentage suppression was calculated based on the CFSE dilution of the target T-cells in the suppression culture, as described[38]. Therefore, the percentage of target cells dividing in response to the stimuli alone, in the presence of expanded Tregs or control effector T-cells, was determined. The percentage suppression was then calculated by the percentage reduction in proliferation of target cells with suppressors compared to target cells alone.

## Statistical analyses

Statistical analyses and cumulative data presentation were performed with Prism 9 (GraphPad Software Inc.). We used the Shapiro–Wilk test for testing of normality. Comparison of the measured values was performed using ANOVA with Holm–Sidak post hoc test or Kruskal Wallis with Dunn's post hoc test depending on the results of Shapiro–Wilk test, with $p < 0.05$ (*), $p < 0.01$ (**), and $p < 0.001$ (***) indicating statistically significant differences. To detect significant differences in multifunctional T-cells, we used Fisher's exact test. The correlation analyses were carried out using the software Jamovi version 2.2.21.

## Reporting summary

Further information on research design is available in the Nature Portfolio Reporting Summary linked to this article.

## Data availability

The mRNA sequencing data that has been generated in this study is deposited in the Gene Expression Omnibus database entitled "mRNAseq profiling of *Bifidobacterium longum*-stimulated neonatal and adult T-cells" under accession code GSE210336. Source data are provided in this paper.

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

## Acknowledgements

The authors would like to thank Kathrin Kramer and Annette Sohnekind for excellent technical support and the midwives of Marienstift for support. The study was founded by the Deutsche Forschungsgemeinschaft (DFG, German Research Foundation) Project-ID Br1860/12 (MB-W) and TRR 359—Project number 491676693 (D Bruder), by the BMBF (LongCoCID, Project-ID 01EP2101C; MB-W) and by the ministry of Saxony-Anhalt (Project-ID I 196; MB-W).

## Author contributions

K.V. and M.B.-W. designed the study. K.V. planned the work, performed the experiments, interpreted the results, and co-wrote the paper with M.B.-W. and A.A. D.S., M.Z. and R.G. performed RNASeq assays and analysis, participated in the interpretation of the findings, read, and critically revised the paper. S.B., D. Bretschneider, F.P., D. Bruder, T.H. and C.A. contributed reagents and materials and provided expertise and feedback. H.L. participated in data analysis and revised the paper. M.B.-W. secured funding and supervised the study. All co-authors read the paper and have provided important intellectual input to the paper. The corresponding author had full access to the data and had final responsibility for the decision to submit it for publication.

## Funding

## Competing interests

The authors declare no competing interests.
