## [Peer Review File · Nature Communications]

Bifidobacteria shape antimicrobial T-helper cell responses during infancy and adulthoodREVIEWER COMMENTS

Reviewer #1 (Remarks to the Author):

In this study, Vogel and colleagues use in vitro assays of human CD4+ T cells to assess the effect of microbial-derived peptide exposure on T cell activation and differentiation outcomes. T cells have been isolated from umbilical cord vein after parturition or infant adenoids and compared with those from adult donor blood, as it has been previously shown that T cell activity differs in early life. T cell responses against microbe exposure was compared between *B. longum*, an early colonizing bacteria obtained from mother's milk, and *S. aureus* and *S. epidermidis*, commensal bacteria with the potential for pathogenicity. This is accomplished by culturing CD4+ T cells, either as a population or sorted into RTE, naïve, or memory subsets, with human monocytes that were previously incubated with heat-treated lysates from the respective bacterium, and assessing proliferation (by CFSE dilution), activation (by upregulation of CD25), and/or Th effector cytokines or markers. Unlike *B. longum*, *S. aureus/epidermidis* elicits Th1 response in T cells of all ages, with particular upregulation of inflammation in neonatal T cells. The authors hypothesize that neonatal T cells are disposed to adopt a suppressive Treg signature in response to *B. longum*, which protects against hyperactivation against *S. aureus*, a protection that is dependent on CTLA-4. Relevant to current immunology, CD4+ T cells isolated from COVID-19 patients were treated with SARS-CoV-2 peptides and found to have reduced Th1 activation and increased IL-10 production when also exposed to *B. longum*.

The topic of the study is interesting, and the focus on human cells increases its significance. The work is highly descriptive, and the reviewer is uncertain whether it meets the level of impact for the publication. I kept waiting for a model, even proposed, as to why neonatal T cells have different reactivity thresholds, or why certain bacteria evoke immunosuppressive programs. A compelling piece of evidence would be to use neonatal T cells that have been exposed to *B. longum* in a suppression assay, to show that they are indeed functioning as claimed. It is interesting that neonatal T cells stimulated with *B. longum* generate more IL-10, but that dampened T cell response with CTLA-4 dependent not IL-10 dependent, it would be good to see if the *B. longum*-stimulated T cells needed to be able to contact the *S. aureus*-stimulated T cells in order to have an effect. Copyediting of the manuscript is also encouraged to improve readability and adjustments to the figures for clarity. I submit my comments and recommendations below:

- Abstract: a sentence to introduce the bacterial species employed would improve the logic behind the results; also, a conclusion as to what this study will inform us about the differences between T cells generated in early life and adulthood.
- P.4 line 83: "...correlated with accumulation of Treg" did this study show that Treg are induced by *B. longum*, or that they are stimulated?
- P. 6 line 132: "...T cell proliferation was unambiguously detectable by CFSE dilution upon stimulation with... h. i. *B. longum*" not sure if really unambiguous, it is hard to see the proliferating group. Perhaps a xy plot for Figure 1D?
- P. 7 line 139: "Correlating... indeed age dependent" is there a statistical method to show this is a true correlation? Hard to tell from graph 1E.
- P. 7 line 162: "... had more IL-2 producers in neonates than..." have to eyeball from the graphs, could there be a statistical comparison of the fold difference?
- P. 8 line 173-175: "Bacteria-specific cytokine producers..." unable to understand what this sentence is trying to say
- P. 8 line 195-197: "...CD25 expression and abundant cluster formation but low response..." not following the logic, because it seems that CD25 expression is still not that great for *B. longum*, so why conclude that low proliferation and cytokine production is not just a result of reduced activation?
- P. 9 line 199: "...antibodies against HLA-DR." what is the rationale for doing this? I presume to show that the effect is TCR-mediated, but again, CD25 upregulation does not seem that great anyway.
- P. 9 line 206: could supplement table of the 18 molecule list that was compiled?
- P. 9 line 214: could you explain what galectin-1 is and why this difference between neonatal and adult could be important?
- P. 9 line 224: not sure if the GO analysis is with regards to neonatal or adult T cells, and why does oxidative metabolism mean that they are Treg?
- Please set the supplementary data so that they are presented in order of the text.
- Figure 1A: the x axis as human months is not intuitive to read.
- Figure 2: the key at the bottom of G has the conditions out of order, please put *B. longum* last

since it is the last group in each chart.

- Figure 4A. The key says that this panel is color-coded, but it is not.
- Figure 4F. I did not see Discussion as to why the capsid peptide + *B. longum* treated cells have greatly increased IL-10. Also, presumably the COVID patients are adults? And these are bulk CD4 T cells? Need to discuss how comparable these experiments are to the earlier ones.

Reviewer #2 (Remarks to the Author):

The paper by Weinzierl et al addresses the Trajectory and Magnitude of Antimicrobial T-helper Cell Responses during Infancy and Adulthood in response to in vitro stimulation of PBMC with 3 different bacteria.

The authors conclude that *Bifidobacterium longum* induces a regulatory T cell phenotype whilst *Staph aureus* polarises the T cell response towards a Th-1 spectrum, and that these responses are age dependent.

Why the authors decided to include a few observations in patients with severe COVID is not clear as it does not sit well within the narrative of the manuscript and shift the focus entirely into the immunomodulatory role of *B.longum* rather than pursuing the narrative of age-and pathogen dependent responses.

Introduction:

The underlying hypothesis for this series of experiments is not entirely clear- T cell responses are not the first line of defense against bacterial infections. This is not mentioned. The concluding 2 sentences of the introduction are purely speculative and don't add at this stage. Instead the authors should state a clear hypothesis for their work.

Methodology:

There are several open questions which have implications for the methodology and subsequently the results:

Host cells:

Where were the samples derived from? There is no information on the neonates or children or adults with Sars-CoV-2, and baseline demographics such as age, sex, disease severity and the like need to be included.

Were these neonates born by vaginal delivery or caesarean section? Were they term infants?

The numbers of samples from infants in each age category which go forward to the in depth phenotyping are very small- when initially a large number of samples appears to have been available for the bacterial stimulation experiments-please explain the reason/selection

Bacterial stimulations:

Why the choice of the 3 organism (*Staph aureus*, *Staph epi*, *B.alongum*) when one is a commensal gut bacterium and the other 2 primarily skin commensals with potential to cause invasive disease?

Where are the concentration and timecourse experiments, and why where these specific -pretty high- MOI's chosen. Were the antigen concentrations comparable? What is the LPS content for each of these bacteria as non-specific responses and CD25 expression can be expected, especially in *Staph aureus*. Was this a toxin-producing *staph aureus*? Was this issue excluded?

Results:

The stimulation experiments work nicely, also positive control, so the basic set up is fine, except might be time course and antigen concentration dependent-see methods comment.

It is surprising though that the responses to the positive control do not appear age dependent- we

usually see lower responses to non-specific stimuli such as PHA in younger children-the the curve looks flat?

T cell activation and proliferation are clearly demonstrated-how were the different numbers of naïve T cells in circulation between newborns, younger and older children taken into account when interpreting the responses?

Cytokine responses are known to be age dependent- IFN γ promoter maturation is delayed in the first 5 years of life- this could be picked up in the discussion.

Not all TLR signalling depends on MYD-88 hence it would be necessary to see additional blockers as TLR signalling will play a crucial role in immune activation.

Fig 2 G-H only reports on the B Longum responses- what happened with the other 2 bacteria?

"early in life half of the induced FoxP3+ Tregs expressed FoxP3 without GATA3, T-bet or ROR γ t"- where do the authors think, the FOX-P3 is coming from then?

Inclusion of the e Sars-CoV2 patients seems opportunistic and shifts the focus away from trying to understand the responses to different bacteria and age impact and b.longum becomes the main focus- this is a different narrative in my view and does not help the manuscript. The numbers of samples included here is extremely small, there is no detail on the background of samples, disease phenotype etc., hence these findings are not conclusive-yet a lot of emphasis is put on it at the end of the discussion.

Discussion:

I take issue with this statement: "These extremely different responses reflect well the immunological requirements in the first years of life, when immunological memory cannot yet be invoked" Vaccines in early infancy can induce entirely potent memory responses- this is the reason why they work 9

Line 294: grammar?? Word missing?

Line 319: what exactly are the authors saying here? Uninterpretable

It sounds as if the b longum responses are not SARS-Cov-2 specific-if indeed they exist- but also apply to influenza- this then raises a more comprehensive picture of potential interference in type-1 interferon pathways? Needs to be discussed if indeed this issue is to be pursued in a bit more depth.

In my view there is a huge jump at the end of the discussion to adult data from very few non-characterised COVID patients. The sequence of bacterial exposure and subsequent immune modulation is an interesting subject but not sufficiently addressed experimentally in this work to merit such far reaching conclusions right at the end.

Minor issues:

there are a number of stylistic and grammatical errors and some labelling issues: mislabelling in the legends/graphs, e.g. Fig 2 D & E appear switched?

Fig 3: isolated from

Fig 4: all of a sudden called corona patients?

Reviewer #3 (Remarks to the Author):

Thank you for inviting me to review the manuscript. The authors describe the effects of several relevant bacteria on the responses and development of T-cells obtained from individuals across the lifespan.

The manuscript is well written and the topic is quite relevant.

I have several comments that I hope will improve the manuscript. I have two major comments and some other minor comments that I believe the authors can easily address.

Major points:

1. Line 77-78: *B. infantis* is actually very rare among infants in the EU and USA (for example, see PMIDs: 27133167 and 33479326) and is enriched only in a limited number of communities (e.g., PMID:33905556). It would be more accurate to note that this organism is capable of colonizing the infant gut early in life (rather than noting that it 'is one of the earliest colonizers') (cite PMID:19033196) and outcompeting *Staphylococcus* and *Streptococcus* (cite PMID:29242832), decreases inflammation (cite PMID:34143954) and noting that its growth is supported by components found in human milk (instead of "provided by mother's milk") (PMID 21342711).

The citations noted don't fully support the (correct) statement that *B. infantis* out-competes other potentially pathogenic/inflammatory taxa, but adding the recommended citations above and adjusting the text will improve the accuracy of the statement.

2. Line 383: A major question I have about this work is how the cells were heat inactivated. It is not clear from the manuscript, but excessive heating might remove some of the volatile metabolites produced by the cells. Further, were the cells washed before heat inactivation? Please carefully describe the methods for the generation of the cells used here.

Minor comments:

1. The use of *B. longum* throughout is a little confusing (e.g., Line 77) when referring to *B. infantis*. While this is the species name, the subspecies is most commonly used to refer to the organism in a shortened form after the first usage. Given the phenotypic distinction between *B. longum* subsp. *longum* and *B. longum* subsp. *infantis*, it seems most appropriate to use *B. infantis* throughout.

For example, on the first usage use: "...*Bifidobacterium longum* subsp. *infantis* (*B. infantis*)..."

2. It is unclear from the manuscript what the number of donors was or any demographics about them (e.g., age). It would be helpful to include this information, maybe as a table, to note the age and number (and other relevant characteristics) of the participants.

3. Line 202-219: I think the authors might mean genes as this is RNASeq data being described. Or, if it's a number of classes of compounds, list them in a table with their respective references?

4. Line 227-228: "By analyzing..." is a sentence fragment.

5. Line 234: "parameters"

6. Line 251: include a comma after "early in life" for clarity

7. Line 278: It's unclear what the authors are referencing by "peptides pulsed monocytes to..."

8. Line 351: The authors should soften this statement. As there is not an identified mechanism for the interactions observed with *Bifidobacterium*, it is premature to broadly reference the use of any *Bifidobacterium* as other strains or species may not act in the same way. Please soften this statement, refer specifically to *B. infantis*, and/or add a note cautioning that the mechanism for the observed functions is not known.

9. Line 352: "corona patients" needs to be revised.

18th of May 2023

Dear reviewers,

Please find below our point-by-point response to the reviewers' comments and the revised manuscript entitled "Bifidobacteria shape Trajectory and Magnitude of Antimicrobial T-helper Cell Responses during Infancy and Adulthood" by Vogel et al. and Brunner-Weinzierl, submitted for publication as an original research article in Nature Communications.

We have carefully read the comments and recommendations of the reviewers and believe that all concerns raised can be addressed experimentally or by revising the manuscript. Based on the reviewers' suggestions, we have performed new experiments to validate our key observations. Below we provide a point-by-point response to the reviewers' comments along with a revised version of the manuscript. We hope that we have satisfactorily addressed all concerns, particularly those related to study design, sample demographics and methodology.

We thank you for your kind attention and hope that you will find our manuscript worthy of consideration.

Yours sincerely,

Monika Brunner-Weinzierl

REVIEWER COMMENTS

Reviewer #1:

We thank the reviewer 1 for the valuable comments and recommendations and we hope that the revised version now meets the level of impact for the publication.

“In this study, Vogel and colleagues use in vitro assays of human CD4+ T cells to assess the effect of microbial-derived peptide exposure on T cell activation and differentiation outcomes. T cells have been isolated from umbilical cord vein after parturition or infant adenoids and compared with those from adult donor blood, as it has been previously shown that T cell activity differs in early life. T cell responses against microbe exposure was compared between *B. longum*, an early colonizing bacteria obtained from mother’s milk, and *S. aureus* and *S. epidermidis*, commensal bacteria with the potential for pathogenicity. This is accomplished by culturing CD4+ T cells, either as a population or sorted into RTE, naïve, or memory subsets, with human monocytes that were previously incubated with heat-treated lysates from the respective bacterium, and assessing proliferation (by CFSE dilution), activation (by upregulation of CD25), and/or Th effector cytokines or markers. Unlike *B. longum*, *S. aureus/epidermidis* elicits Th1 response in T cells of all ages, with particular upregulation of inflammation in neonatal T cells. The authors hypothesize that neonatal T cells are disposed to adopt a suppressive Treg signature in response to *B. longum*, which protects against hyperactivation against *S. aureus*, a protection that is dependent on CTLA-4. Relevant to current immunology, CD4+ T cells isolated from COVID-19 patients were treated with SARS-CoV-2 peptides and found to have reduced Th1 activation and increased IL-10 production when also exposed to *B. longum*.”

1. The topic of the study is interesting, and the focus on human cells increases its significance. The work is highly descriptive, and the reviewer is uncertain whether it meets the level of impact for the publication. I kept waiting for a model, even proposed, as to why neonatal T cells have different reactivity thresholds, or why certain bacteria evoke immunosuppressive programs. A compelling piece of evidence would be to use neonatal T cells that have been exposed to *B. longum* in a suppression assay, to show that they are indeed functioning as claimed.

We included models and explanations of the two issues, namely altered activation threshold of neonatal T cells and why do some bacteria evoke immunosuppressive programs. For that, please see extensive rewriting at the beginning of the discussion section.

We thank the referee for the suggestion to strengthen our results about the suppressive functioning of the *B. infantis*-specific T cells. We performed a suppression assay and indeed found strong suppressive functioning in suppression assays (see new Fig. 5 in the main body of the manuscript and below).

Fig. 5D in the manuscript: T cells exposed to *B.infantis* tested in a standard suppression assay. Neonatal and adult $CD4^+CD45RA^+CD31^+$ T-cells were co-cultured for 3 days with monocytes matured with h.i. *B. infantis*, and assayed for suppressive activity using CFSE-labeled resting target T cells and different ratios of T-cells stimulated prior assay onset for 3 days with *B. infantis*-matured monocytes. The mix of cells were then stimulated with anti-CD3 antibody in the presence of dendritic cells for 4 days, and proliferating cells were determined based on decrease in CFSE fluorescence. Percentages of suppression were calculated based on CFSE dilution of target T-cells in suppression culture. Cumulative results are shown and each dot represents one donor. Correlation analysis was performed using Spearman's Rho, with *** representing $p < 0.001$. Data shown are representative of at least 5 experiments performed with T-cells from different donors. The lines show a linear approximation of the logarithmically transformed ratio of T suppressor cells to target cells.

2. It is interesting that neonatal T cells stimulated with *B. longum* generate more IL-10, but that dampened T cell response with CTLA-4 dependent not IL-10 dependent, it would be good to see if the *B. longum*-stimulated T cells needed to be able to contact the *S. aureus*-stimulated T cells in order to have an effect.

Since the inhibition of *S. aureus*-stimulated T cells could be reversed by blocking the surface-expressed molecule CTLA-4, it is likely that cell-cell contact is required. To provide further evidence for the necessity of cell contacts, we now took an additional approach of using trans-well chambers (see below and new Fig. 5B). Both stimulations were performed in parallel, separated by a net. Inhibition was not detectable, reinforcing the results that the suppressive capacity of *B. infantis* exposed T cells requires cell-cell contact with target cells.

Fig. 5C of the manuscript: T-cells were stimulated in trans-well-plates that differently stimulated cells did not get in contact, but used the same media. (C) Statistical analysis of Transwell assays with purified naïve (CD4⁺CD45RA⁺CD31⁺) T-cells of neonates and adults. Thereby, the content of the upper chamber is indicated above the line and the content of the lower chamber is indicated below the line. T-cells in the lower chamber were labelled with CFSE, rested (R), or stimulated either with *B.infantis* (B)- or *S.aureus*(S)-matured monocytes in culture for 6 days and analyzed by flow cytometry. Cumulative results are shown and each dot represents a donor. Error bars in figures denote \pm SD. * $p<0.05$, ** $p<0.01$, *** $p<0.001$, **** $p<0.0001$ as determined by one-way ANOVA with Holm-Sidak post hoc test.

3. Copyediting of the manuscript is also encouraged to improve readability and adjustments to the figures for clarity.

Copyediting was done for clarity.

I submit my comments and recommendations below:

- Abstract: a sentence to introduce the bacterial species employed would improve the logic behind the results; also, a conclusion as to what this study will inform us about the differences between T cells generated in early life and adulthood.

In the abstract, bacterial species are introduced and a concluding sentence has been added, now: "Targeted, age-specific interventions may enhance infection defence, and specific immune features may have potential cross-age utilization."

- P.4 line 83: "...correlated with accumulation of Treg" did this study show that Treg are induced by *B. longum*, or that they are stimulated?

The publication *Lyons et al., 2010* shows elegantly in a mouse model increased numbers of Treg cells upon *B. longum* ingestion. These cells are able to suppress other T cells. Nevertheless, it is hard to tell whether these cells are induced (or migrated) or stimulated in an *in vivo* model (which is more the advantage of our setting). Therefore, we avoid induction, but changed the term expression into "...correlated with increased numbers of Treg".

Of note, our model starts with a setup of the CD4 T cell population, each containing depleting antiCD25 antibodies. Since Tregs, including nTreg (tTreg), express CD25, it can be assumed that existing Treg cells were not present in the stimulation (Kit for the isolation of naïve CD4⁺ T cells or the CD4⁺ Recent Thymic Emigrant Isolation Kit (Miltenyi Biotec)). This

leaves to conclude that Treg cells were indeed induced after *B. infantis* stimulation. However, it does not exclude that precursors exist, that have a bias for becoming iTregs (pTregs).

- P. 6 line 132: "...T cell proliferation was unambiguously detectable by CFSE dilution upon stimulation with... h. i. *B. longum*" not sure if really unambiguous, it is hard to see the proliferating group. Perhaps a xy plot for Figure 1D?

We explored different ways to present the data, unfortunately, there is hardly a better way to show the proliferating T cells upon *B. infantis*/APC encounter. Cell detection was tightly controlled, even by MHCII blockade and repeated several times. By far more than 50 T cells were detected to proliferate, the threshold number to consider results positive. In human T cells assays from individuals, we often work with low numbers.

4. - P. 7 line 139: "Correlating... indeed age dependent" is there a statistical method to show this is a true correlation? Hard to tell from graph 1E.

For illustrative purposes, data were previously presented descriptively. Since the data are metric, we have now applied Pearson's correlation in order to statistically validate at least the linear part of the data. Now r and the significance level are inserted with asterisks directly in the graph at the end of the lines of the respective pathogen and polyclonal stimulation across age. Graph 1E is now enlarged for better visualization. The assumption that this is a logarithmic relation has not improved the statistics. Spearman's Rho was also at a similar r and significance level. As shown, the proliferation data are very strongly correlated with age with $p < 0.005$ for each pathogen.

New Fig. 1E of the manuscript: Bacteria-specific T-cell activation. (E) Purified naïve

(CD4⁺CD45RA⁺CD31⁺) T-cells from neonates and adults were labelled with CFSE. Frequency of proliferating (CFSE^{lo}) T-cells from neonates,

infants, children, and adults stimulated for 3 days with h.i. *S. aureus* (blue), h.i. *S. epidermidis* (green), h.i. *B. infantis* (orange) or anti-CD3/CD28 (black), determined by flow cytometry and plotted against age of donor. Each dot represents a different donor. Correlation analyses are carried out with the Pearson's correlation. The Pearson's correlation coefficient r is displayed with *** representing $p < 0.001$.

We added to the text: "Correlating the bacterial antigen-mediated proliferation of T-cells with the age of donors revealed pathogen-specificity and demonstrated age dependency, as shown by Pearson's correlation coefficient ($r < -0.67$, with all p -values < 0.001) and Spearman's Rho ($Rho < -0.86$, with significances) (see Fig. 1E)."

5. - P. 7 line 162: "... had more IL-2 producers in neonates than..." have to eyeball from the graphs, could there be a statistical comparison of the fold difference?

Statistics are shown on top of the bars (for *Bifido*-responses in yellow-brown). In addition, values and statistics can now also be seen in numbers in tables in the appendix. Thank you for critical reading and we apologize for the mistake in the text. We changed it according to the statistics. Indeed, neonates, infants, and children show higher frequencies of IL-2 producers than adults do. We now corrected the sentence:

"In contrast, even though at low frequencies *B. longum* stimulated T-cells had significantly more IL-2 producers in neonates, infants and children than in adults (Fig. 2A,B, left)."

6. - P. 8 line 173-175: "Bacteria-specific cytokine producers..." unable to understand what this sentence is trying to say

We apologize for not being clear enough and changed the sentence, which is referring to Bob Seders model about multifunctional T-cells:

"Next, T cells, which express several cytokines simultaneously and are thus considered particularly potent effector T cells, were examined after bacteria-specific stimulation."

7. - P. 8 line 195-197: "...CD25 expression and abundant cluster formation but low response..." not following the logic, because it seems that CD25 expression is still not that great for *B. longum*, so why conclude that low proliferation and cytokine production is not just a result of reduced activation? - P. 9 line 199: "...antibodies against HLA-DR." what is the rationale for doing this? I presume to show that the effect is TCR-mediated, but again, CD25 upregulation does not seem that great anyway.

The main argument for the recognition of *B. infantis* by T cells and their activation is concluded from our data showing that the activation-induced clusters are as numerous as compared to *S. aureus* induced T-cell activation. These clusters only occur when the T cells are activated (Fig. 1B,C).

The frequency of CD25⁺ (8-10% of CD4 T cells) and proliferating T cells is still quite high when inexperienced T cells are activated antigen-specifically in the human system, considering TCR diversity. The frequency is significantly ($p < 0.005$) higher compared to resting T cells, showing that activation is taking place. We need to learn more about these cells to obtain even better surrogate markers for their antigen-specific activation. At this stage, we assume that the high and stable clustering show that the *B. infantis*-specific T cells recognize and respond to *B. infantis*.

Therefore, we feel like the story needs the data to give a complete picture. We changed the sentences to make our point clear:

"Next, we investigated the mechanism behind *B. infantis*-specific T-cell activation as determined mainly by abundant cluster formation (Fig. 1), though low (still about 10% responding cells in a polyclonal setting), but significantly positive, response in terms of frequencies of CD25-expressing cells and proliferation or cytokine production (Fig. 1-2)."

8. - P. 9 line 206: could supplement table of the 18 molecule list that was compiled?

The 18 molecules are now added as information in the supplemental Tab. S15 (see below).

Table S15. Genes analysed by RNA-Seq			
Name	ENSEMBLE_ID	Gene	Description
FoxP3	ENSG00000049768	AIID, DIETER, IPEX, JM2, PIDX, SCURFIN, XPID	forkhead box P3
IL-10	ENSG00000136634	CSIF, IL-10, IL10A, TGIF	interleukin 10
TGFbeta	ENSG00000105329	TGFB1	transforming growth factor beta 1
Galectin-1	ENSG00000100097	LGALS1	galectin 1
CD25	ENSG00000134460	IL2RA	interleukin 2 receptor subunit alpha
CTLA-4	ENSG00000163599	CTLA4	cytotoxic T-lymphocyte associated protein 4
IL35B	ENSG00000105246	EBI3, IL27B	Epstein-Barr virus induced 3
GITR	ENSMUSG00000041954	Tnfrsf18	tumor necrosis factor receptor superfamily, member 18
LAG-3 (CD223)	ENSG00000089692	LAG3, CD223	lymphocyte activating 3
CD39	ENSG00000138185	ENTPD1, ATPDase, CD39, NTPDase-1, SPG64	ectonucleoside triphosphate diphosphohydrolase 1
IL7R (CD127)	ENSG00000168685	IL7R	interleukin 7 receptor
CD73	ENSG00000135318	NT5E, CALJA, CD73, NT5, eN, eNT	5'-nucleotidase ecto
Helios	ENSG00000030419	IKZF2, Helios, ZNFN1A2	IKAROS family zinc finger 2
STAT5A	ENSG00000126561	MGF, STAT5, STAT5A	signal transducer and activator of transcription 5A
BDCA4	ENSG00000099250	NRP1; CD304, NRP, VEGF165R	neuropilin 1
PD-1	ENSG00000188389	PDCD1, CD279, PD-1, PD1, SLEB2, hSLE1	programmed cell death 1
CD137	ENSG00000125657	TNFSF9, 4-1BB-L, 4-1BBL, CD137L	TNF superfamily member 9
CD103	ENSMUSG00000005947	Itgae, CD103, alpha-E1	integrin subunit alpha E, epithelial-associated

9. - P. 9 line 214: could you explain what galectin-1 is and why this difference between neonatal and adult could be important?

Galectin-1 has been shown to play an important role in the regulation of T-cell responses. Studies have shown that galectin-1 can promote the expansion and survival of regulatory T cells (Tregs) and enhance their immunosuppressive function (1,2). Galectin-1 has also been shown to promote the differentiation of Tregs from naive T cells, suggesting that it may be involved in the development of immune tolerance. In humans, a deficiency of bifidobacteria and/or their genes required for the utilization of human milk oligosaccharides from breast milk is associated with systemic inflammation and immune dysregulation early in life. *Bifidobacterium infantis* has been shown to reduce Th2 and Th17 cytokines in the gut by upregulating the immunoregulatory factor galectin-1 (3).

The enhanced expression in neonatal response goes well with inhibitory mechanisms needed during this critical period of life as galectin-1as in neonates, Tregs are particularly important for establishing immune tolerance and preventing immune-mediated damage to developing tissues.

References showing involvement of galectin-1 and inhibition of T-cell responses:

1. Toscano MA, et al. Galectin-1 suppresses autoimmune retinal disease by promoting concomitant Th2- and T regulatory-mediated anti-inflammatory responses. 2006. J. Immunol.:176:6323.
2. Garín, M. I. et al. Galectin-1: a key effector of regulation mediated by CD4+CD25+ T cells. 2007. Blood 109, 2058.

3. Henrick, B. M. et al. Bifidobacteria-mediated immune system imprinting early in life. *Cell* 184, 3884-3898.e11; 10.1016/j.cell.2021.05.030 (2021).

We have accordingly added this information to the results and discussion of the manuscript.

10. - P. 9 line 224: not sure if the GO analysis is with regards to neonatal or adult T cells,

We apologize for not being clear enough. The GO analysis was done with neonatal T cells shown in the supplemental of the manuscript. We clarified it in the main body of the manuscript and in the figure legend by adding neonatal to the head line.

11. and why does oxidative metabolism mean that they are Treg?

Treg cells have been suggested to have a distinct metabolic profile that is characterized by a high level of oxidative metabolism. This metabolic profile is in contrast to the glycolytic metabolism that is typical of activated T cells, which undergo rapid proliferation and require a lot of energy. Treg cells rely on oxidative metabolism to generate energy and maintain their viability, and this metabolic profile is critical for their ability to regulate immune responses and maintain immune system homeostasis. Furthermore, studies have shown that Treg cells have a higher expression of genes involved in oxidative phosphorylation and fatty acid oxidation, and a lower expression of genes involved in glycolysis, compared to effector T cells (Guglielmetti *et al.*, 2019; Wang *et al.*, 2021). In addition, Sun and colleagues suggested that Treg cells are involved in neonatal T cells responding to *B. ssp. longum* and used GO and genes of oxidative phosphorylation as an argument that Tregs are involved (Sun *et al.*, 2020).

References:

Guglielmetti, S., Serrano, D., de la Cuesta-Zuluaga, J., et al. "High Level of Oxidative Metabolism Defines the Metabolic Profile of Human Regulatory T Cells." *Nature Communications* 10, 5512 (2019). doi: 10.1038/s41467-019-13321-9 .

Wang, R., Green, D.R., and Metabolism, T.C. "Oxidative phosphorylation in T cell activation and differentiation." *Nature Reviews Immunology* 21, 599-614 (2021). doi: 10.1038/s41577-021-00530-w

Sun, S. et al. Bifidobacterium alters the gut microbiota and modulates the functional metabolism of T regulatory cells in the context of immune checkpoint blockade. *PNAS* 117, 27509-27515 (2020).

12. - Please set the supplementary data so that they are presented in order of the text.

Supplemental data has now been sorted in the same order as it fits the text

13. - Figure 1A: the x axis as human months is not intuitive to read.

Labelling of the x axis was changed from months into years.

14. - Figure 2: the key at the bottom of G has the conditions out of order, please put *B. longum* last since it is the last group in each chart.

Order of the key in Figure 2G has been changed accordingly.

15. - Figure 4A. The key says that this panel is color-coded, but it is not.

Thank you. It has been corrected in the figure legend of Fig. 4.

16. - Figure 4F. I did not see Discussion as to why the capsid peptide + *B. longum* treated cells have greatly increased IL-10. Also, presumably the COVID patients are adults? And these are bulk CD4 T cells? Need to discuss how comparable these experiments are to the earlier ones.

To verify this surprising finding, we have performed more experiments and improved significance of the data with increased IL-10 in Fig. 4F.

Yes, that is right. T cells of severe COVID-19 patients are bulk CD4 T cells from adults which we have clarified in the text, now. In the light that *B. infantis* suppresses Th1-producers (see panel next to it), nucleocapsid stimulation of the T cells at first might result in an enhanced Th2 response when *B. infantis*-specific T cells have been added after 3 days - mirrored by increased numbers of IL-10 producers as detected by ELISpot. In the case of strong suppression of a response - when Th1 is fully suppressed as well as proliferation, because *B. infantis* is first in the stimulation - likely also Th2-responses are suppressed leading to less IL-10.

In Figure 3D, adult T cells show significantly increased IL-10 concentrations in the supernatant upon stimulation with *B. infantis*. We agree that it is a different set up using bulk CD4⁺ T cells. However, it goes well with data in Fig. 3F, where IL-10 producers appear upon stimulation with *B. infantis* and therefore extends our findings to other set ups. The comparability issue has been addressed in the manuscript.

Reviewer #2 (Remarks to the Author):

The paper by Weinzierl et al addresses the Trajectory and Magnitude of Antimicrobial T-helper Cell Responses during Infancy and Adulthood in response to in vitro stimulation of PBMC with 3 different bacteria.

The authors conclude that *Bifidobacterium longum* induces a regulatory T cell phenotype whilst *Staph aureus* polarises the T cell response towards a Th-1 spectrum, and that these responses are age dependent.

1. Why the authors decided to include a few observations in patients with severe COVID is not clear as it does not sit well within the narrative of the manuscript and shift the focus entirely into the immunomodulatory role of *B.longum* rather than pursuing the narrative of age-and pathogen dependent responses.

We know what the reviewer means, however, we are interested since years in inhibitory mechanisms. Therefore, we were looking for age-characteristic bacterial T-cell responses, but also for inhibitory mechanisms and found *B. infantis*-stimulation of T cells. We agree that at the end of the manuscript, *B. infantis* is the major issue. However, it is only a model for an overreactive response in addition that antigens for the T cells are known. Therefore, it is a suitable model. Nevertheless, we reduced the discussion about SARS-CoV-2.

2. Introduction: The underlying hypothesis for this series of experiments is not entirely clear- T cell responses are not the first line of defense against bacterial infections. This is not mentioned. The concluding 2 sentences of the introduction are purely speculative and don't add at this stage. Instead the authors should state a clear hypothesis for their work.

Although it is well recognized that the immune system of neonates and infants differs from that of adults, not much is known about its ability to respond specifically to bacterial antigens such as those of *Staphylococcus* and *Bifidobacterium*. These bacteria are of particular interest because newborns are immediately and continuously exposed to them throughout their lifetime. Therefore, it is expected that there has been co-evolution of this human-microbe interaction that has shaped the human adaptive immune response. We hypothesize that antigen-specific pro- and anti-inflammatory T cell responses to abundant bacteria are formed very early in life in an age-specific manner.

3. Methodology:

There are several open questions which have implications for the methodology and subsequently the results:

3.1 Host cells:

Where were the samples derived from? There is no information on the neonates or children or adults with Sars-CoV-2, and baseline demographics such as age, sex, disease severity and the like need to be included.

We apologize for the missing demographical information about neonates and children as well as SARS-CoV-2 patients, which we have now enclosed in tabular form. These donors have been characterized extensively. Please, see Tab. S1-3.

Were these neonates born by vaginal delivery or caesarean section? Were they term infants?

Please, find all demographical data in Tab. S1-3. The neonates were all born mature (39th-41st week, GA), most of them vaginally (Tab. S1 and below).

Table S1. Characteristics of neonatal research subjects and their mothers.		
Characteristics	Neonates (n=23)	Mothers (n=23)
Gestational age – week of gestation (range)	39.3±1.3 (37-41)	
Sex – no. (%)		
Male	13 (56.5)	
Female	9 (39.1)	
Height – cm	51.9±2.8	
Weight – cm	3596±473	
Caesarean section – no. (%)	6 (26.1)	
Medicinal induction of labour – no. (%)	5 (21.7)	
Age – y. (range)		30±4 (22-37)
Allergy – no. (%)		11 (47.8)
Medication taken during pregnancy – no. (%)		7 (30.4)
Co-morbidities such as – no. (%)		
Disease of the nervous system		2 (8.7)
Metabolic disorder		3 (13)
Blood clotting disorder		1 (4.35)
Drug allergy		2 (8.7)
Acute infection		1 (4.35)
Disease of the gastrointestinal tract		1 (4.35)

The size of each group is indicated in total numbers (no.) as well as percentages (%) in respect to the entire collective. For one neonatal research subject information about gender was not available.

3.2 The numbers of samples from infants in each age category which go forward to the in depth phenotyping are very small- when initially a large number of samples appears to have been available for the bacterial stimulation experiments,-please explain the reason/selection

Unfortunately, often only low amounts of blood is available, which has to be sorted twice before the antigen-specific stimulation and again at the end of the stimulation. As a result, some samples had too few cells after the second sorting. Therefore, originally, 6 samples were taken and prepared for detailed analysis of RNASeq. Only 4 of the remaining 6 enriched RNA samples passed the quality check for sequencing. However, this was discussed with the NGS at the HZI Braunschweig and was considered sufficient to perform the required analyses.

During the pandemic, it became more difficult to obtain samples because operations (adenoidectomies) on small children and infants were drastically reduced due to the need for more nurses in intensive care units. In addition, anxious parents were less willing to give their consent, and the shortage of doctors seems to have reduced their willingness to take the time to inform them. We are also setting up collaborations with other local clinics to increase access to samples from young children. This takes some time because another ethics committee has to be involved. However, the advice from our ethics committee is that we need to stop collecting samples as soon as the data are significant, which we have done.

3.3 Bacterial stimulations:

Why the choice of the 3 organism (Staph aureus, Staph epi, B.alongum) when one is a commensal gut bacterium and the other 2 primarily skin commensals with potential to cause invasive disease?

The three bacteria were chosen because the neonate is in direct contact with them. Therefore, if it occurs at all, an evolutionary symbiosis and coordinated T-cell response to this microbial exposure could have developed. All early colonizing bacteria that can be obtained either from human milk and *S. aureus* and *S. epidermidis* vaginally and/or from the mother's skin. We added this information to the introduction of the manuscript.

3.4 Where are the concentration and time course experiments, and why where these specific -pretty high- MOI's chosen. Were the antigen concentrations comparable?

For the MOIs, the maximum frequency of mediated T cell proliferation was determined and used in further assays. The concentration that resulted in maximal proliferative responses and viability was chosen: $5\mu\text{g ml}^{-1}$ for *S. aureus* and *S. epidermidis* and $10\mu\text{g ml}^{-1}$ for *B. infantis*. See Materials and Methods and Fig. S2 and below. This is in agreement with studies done by Zielinski and Sallusto (Zielinski *et al.*, Nature. 2012 484(7395):514-8. doi: 10.1038/nature1095).

Fig. S2B in the manuscript: Experimental set up as used in Fig.1 and 2 of the manuscript. Proliferation was measured using CFSE labelling and dilution to daughter cells in proliferating T cells in response to bacterial stimulation. T cells were monitored by flow cytometry. Percentage give frequencies of proliferated T cells.

3.5 What is the LPS content for each of these bacteria as non-specific responses and CD25 expression can be expected, especially in Staph aureus.

We apologize for missing information. Endotoxin concentrations were measured using Pierce LAL Chromogenic Endotoxin Quantitation Kit (Thermo Scientific). Usually 1:10 dilution was done to mature monocytes overnight meaning in our cases less than <0.036 EU/ml. Thereafter, it was extensively washed and there should not be any trace left when T cells were added. Therefore, endotoxin levels could be unneglectable.

Of note, for medical devices, using the extraction volume recommendations described below, the limit is 0.5 EU/mL or 20 EU/device for products that directly or indirectly contact the cardiovascular system and lymphatic system. For devices in contact with cerebrospinal fluid, the limit is 0.06 EU/mL or 2.15 EU/device. (FDA Guidance for Industry, Pyrogen and Endotoxins Testing: Questions and Answers, June 2012).

3.6 Was this a toxin-producing staph aureus? Was this issue excluded?

This issue was excluded. The strain used was *Staphylococcus aureus subsp. aureus* Rosenbach (ATCC 25923) that is a “no toxin producing strain”. We added the information to the Materials and Methods section.

4. Results:

The stimulation experiments work nicely, also positive control, so the basic set up is fine, except might be time course and antigen concentration dependent-see methods comment.

We are pleased that the reviewer states that “stimulation experiments work nicely, also positive control, so the basic set up is fine”.

4.1 It is surprising though that the responses to the positive control do not appear age dependent- we usually see lower responses to non-specific stimuli such as PHA in younger children-the curve looks flat?

We now determined the linear part of the correlation between proliferation (CFSE^{low}) and age of the subjects using Pearson’s correlation and found indeed a significant correlation of $r=0.674$ with $p=0.001$ for the non-specific stimulation of naïve T cells (0-60 years) using anti-CD3/CD28-coupled to microspheres. This implicates lower proliferation capacity to non-specific stimulation in younger children.

Fig. 1E of the manuscript: Frequency of proliferating (CFSE^{lo}) T-cells from neonates, infants, children, and adults were stimulated for 3 days with h.i. *S. aureus* (blue), h.i. *S. epidermidis* (green), h.i. *B. infantis* (orange) or anti-CD3/CD28 (black), determined by flow cytometry out of more than 12 experiments and plotted against donor age.

Of note: As positive control, we use strong stimulation with anti-CD3/anti-CD28 coupled to microspheres to show that cells are viable and functional. In addition to the stimulus and concentration, the timing might also be different.

However, in Fig. 2B right, 2D, we show reduced response of (younger) children in terms of IFN- γ production and IFN- γ producers.

T cell activation and proliferation are clearly demonstrated-how were the different numbers of naïve T cells in circulation between newborns, younger and older children taken into account when interpreting the responses?

We think this issue can be neglected because of the following arguments: Our lab is also doing diagnostics for the children's hospital and the reference levels for CD3+CD4+ T cells overlap profoundly between different age groups when using absolute numbers $\mu\text{l ml}^{-1}$ blood. In frequencies, the pool of naïve T cells has been similar frequencies within the CD4 T cells, except of cord blood (Schmiedeberg et al., 2011).

CD3+CD4+:

Newborn	957-3106 $\mu\text{l ml}^{-1}$
< 1 year	1143-4380 $\mu\text{l ml}^{-1}$
1-5 years	511-3774 $\mu\text{l ml}^{-1}$
6-13 years	355-1685 $\mu\text{l ml}^{-1}$

Our human *in vitro* models to study T-cell differentiation and anti-bacterial responses uses similar amounts of naïve T cells and monocytes to start of with. Therefore, their responses can be directly compared. But the most important issue is the local concentration of T cells *in vivo*, which means that likely differences in the circulating amount of naïve T cells will not make such a big difference in the local inflammatory response, especially when accumulated in secondary lymphoid organs were they are primed. Of course, it might be different for neonates as naïve T cells have been found in tissues. Nevertheless, comparing naïve T cells with each other seems reasonable.

Cytokine responses are known to be age dependent- IFN γ promotor maturation is delayed in the first 5 years of life- this could be picked up in the discussion.

Thank you for this suggestion. We added in the discussion section:

From 6 years on, naïve T-cells responses remained similar against *S. aureus* and *S. epidermidis* in showing an increased Th1-like response. As IL-17 is considered to be a signatory cytokine for neonates, our data show that indeed, its preference is antigen-specific and continues while infancy proceeds. Likely, losing the bias for IL-17 responses unequivocally leads to the default Th1-pathway. As T-bet-expressing T cells are as abundant in neonatal T cells as in adult T cells (Fig. 2C, left), whereas IFN γ secretion is lower in neonatal and infant T cells upon antibacterial stimulation (Fig. 2 B, right, Fig. 2D), the availability of the IFN γ promotor may be limited, this is consistent with the notion that the IFN γ promotor has been described to be hyper methylated in human neonatal T cells, and T cell stimulation showed reduced IFN γ production in children up to 12 years of age (White *et al.*, 2002; Vogel *et al.*, 2018). As IFN γ suppresses IL-17-responses, reduced availability of the IFN γ machinery in antibacterial-specific responses likely contributes to the bias towards IL-17 expression (Harrington *et al.*, 2005).

White, G. P., Watt, P. M., Holt, B. J. & Holt, P. G. Differential patterns of methylation of the IFN-gamma promoter at CpG and non-CpG sites underlie differences in IFN-gamma gene expression between human neonatal and adult CD45RO⁺ T cells. *Journal of immunology* (Baltimore, Md. : 1950) 168, 2820–2827; 10.4049/jimmunol.168.6.2820 (2002).

Vogel, Katrin; Pierau, Mandy; Arra, Aditya; Lampe, Karen; Schlueter, Dirk; Arens, Christoph; Brunner-Weinzierl, Monika C. (2018): Developmental induction of human T-cell responses against *Candida albicans* and *Aspergillus fumigatus*. In: *Sci Rep.* 8 (1), S. 16904. DOI: 10.1038/s41598-018-35161-5.

Harrington, L. E. et al. Interleukin 17-producing CD4⁺ effector T cells develop via a lineage distinct from the T helper type 1 and 2 lineages. *Nature immunology* 6, 1123–1132; 10.1038/ni1254 (2005).

Not all TLR signalling depends on MYD-88 hence it would be necessary to see additional blockers as TLR signalling will play a crucial role in immune activation.

We excluded TLR1, 2, 4, 5, 6, and intracellular receptors TLR 7, 8, by blocking with specific antibodies or MyD88 inhibitor. According to the ImmGen database, T cells do not express TLR 3, 8 and 9. In addition, we block HLA to detect antigen-specific responses. All this makes it very unlikely that TLRs are involved, at least in our model. We added this information to the results:

Fig 2 G-H only reports on the *B. Longum* responses- what happened with the other 2 bacteria?

All bacteria are plotted in two bar graphs for clarity. Figure 2G shows *B. infantis* plus *S. aureus* and Figure 2H shows *S. epidermidis*. As this was not clear, we have removed the letter H and now refer to left and right bar graphs. We have also clarified this in the text and legend referring to Figure 2.

“early in life half of the induced FoxP3⁺ Tregs expressed FoxP3 without GATA3, T-bet or RORγt”-where do the authors think, the FOX-P3 is coming from then?

As GATA3 and FoxP3 co-expression is a surrogate marker for Treg stability, we actually wanted to make this point that indeed quite stable Treg cells are generated. Therefore, we changed the sentences accordingly. Of note, induction of expression of FoxP3 likely involved e.g. STAT5 (Burchill *et al.*, 2007, *Jl* 178:280; Yao *et al.*, *Blood* 109:4368, 2007), CREB and others at the very beginning of the stimulation. Of note, enhanced STAT5 expression was found in our RNASeq data of *B. infantis*-stimulated neonatal T cells. Here, we address the point later during differentiation that we actually detect Treg cells with certain propensities (GATA3 is not needed for suppressive functioning) of the cells and GATA3 is used as a surrogate marker for Treg stability.

We changed the sentences in the results section: “Furthermore, early in life, half of the induced FoxP3⁺ Tregs co-expressed FoxP3 with either GATA3, T-bet or RORγt. However, exclusively in case of *S. aureus* stimulation, FoxP3 T-cells are constrained in the naïve T-cell pool of adults (Fig. 3H).

Inclusion of the e Sars-CoV2 patients seems opportunistic and shifts the focus away from trying to understand the responses to different bacteria and age impact and *B. longum* becomes the main focus- this is a different narrative in my view and does not help the manuscript. The numbers of samples included here is extremely small, there is no detail on

the background of samples, disease phenotype etc., hence these findings are not conclusive- yet a lot of emphasis is put on it at the end of the discussion.

We have now included the demographic data of all individuals studied in tabular form, please see Tab. S1-3 – COVID-19 patients are in Tab. S3. We have also added more experiments/patients and could increase significances of the results.

The inclusion of SARS-CoV-2 infected patients shows that our assumption is general and could apply to many situations. For this reason, we would like to include the data. We agree that the discussion is too focused on SARS-CoV-2 and shortened this part in the discussion.

Discussion:

I take issue with this statement: “These extremely different responses reflect well the immunological

291 requirements in the first years of life, when immunological memory cannot yet be invoked”

Vaccines in early infancy can induce entirely potent memory responses- this is the reason why they work.

We agree with the reviewer and accordingly changed the sentence. However, we do not refer to vaccination which is a different case as adjuvants are included with which a very strong and lasting response is forced.

Line 294: grammar?? Word missing?

While rewriting the discussion, this sentence was replaced.

Line 319: what exactly are the authors saying here? Uninterpretable

We corrected the sentences: “Despite the induction of an IL-17A program, we show a high potential of naïve T-cells to express FoxP3, which is the master transcription factor of Tregs. This suggests that during the window in early life when Tregs are preferentially induced, antigen-specific T cell responses against bacteria actually serve as triggers for Treg differentiation.”

It sounds as if the *B. longum* responses are not SARS-Cov-2 specific-if indeed they exist- but also apply to influenza- this then raises a more comprehensive picture of potential interference in type-1 interferon pathways? Needs to be discussed if indeed this issue is to be pursued in a bit more depth.

Here, we report on CD4 T cell responses – also when addressing T cells of COVID-19 patients. Even though IFN-type-1 responses are important, we would need to include the main producers of it, CD8 T cells in the studies. However, we feel like we will stay focused on CD4 T cells. However, recruiting severe COVID-19 patients has become difficult due to the current SARS-CoV-2 mutation, which leads to a less severe disease course. In Fig. 4C, we demonstrate that Th1 producers are reduced when *B. infantis*-differentiated CD4 T cells are in cell-cell contact with *S. aureus* stimulated T cells.

However, to provide further evidence that *B. infantis*-stimulated T cells can act broadly to suppress adverse immune responses, further data have now been obtained showing that *B. infantis* exposed CD4 T cells function to suppress effector T cells in a standard suppression assay (see below and Fig. 5D in the manuscript). Indeed, as stated by the reviewer, previous and new suppression assays demonstrate that Treg cells are antigen-specific induced and exert their suppressive function specific and/or likely also as bystander (Thornton and Shevach, JI 164:183, 2000). This adds another Treg-characteristic to our newly described *B. infantis* specific T cells.

Thornton AM, Shevach EM. Suppressor effector function of CD4+CD25+ immunoregulatory T cells is antigen nonspecific. J Immunol. 2000 Jan 1;164(1):183-90. doi: 10.4049/jimmunol.164.1.183

We added to the discussion:

However, CD8 T cells, the major players in viral infections, were not included in our human model. Nevertheless, it can be speculated that at least a mediated reduction in Th1-cell help may ultimately lead to altered CD8 activity. Bifidobacterium may therefore reduce the severity of COVID-19⁵⁷. Whether this is a general mechanism for Bifidobacterium-mediated protection against viral infections, as has been suggested for influenza virus, needs to be pursued further⁵⁸. However, the evidence provided here that *B. infantis*-stimulated T cells can act broadly to suppress adverse immune responses, such as Th cells against *S. aureus*, against *S. epidermidis*, against Sars-CoV2, but most importantly, in a standard suppression assay (Fig.5D) supports this notion. Just as known for tTreg (nTreg) cells, *B. infantis*-specific T cells are antigen-specific induced and exert their suppressive function in a specific and/or bystander manner⁵⁹ (see discussion).

In my view there is a huge jump at the end of the discussion to adult data from very few non-characterised COVID patients. The sequence of bacterial exposure and subsequent immune modulation is an interesting subject but not sufficiently addressed experimentally in this work to merit such far reaching conclusions right at the end.

We were able to add two more patients to the study. The five COVID-19-patients of our intensive care department are well characterized and the information is given in Tab S3, now. We apologized for not providing that information in the first place. We draw back our conclusion to some extent. However, as more suppression assays are now performed, we have much more evidence and therefore feel that now experimental proves are given (see new Fig. 5). Previous and new suppression assays suggest now that our Treg-like cells are antigen-specific induced and exert their suppressive function specific and/or likely also as bystander (Thornton and Shevach, JI 164:183, 2000). This adds another Treg-characteristic to our newly described *B. infantis*-specific T cells.

Fig. 5D in the manuscript: T cells exposed to *B. infantis* tested in a standard suppression assay. Neonatal and adult CD4⁺CD45RA⁺CD31⁺ T-cells were co-cultured for 3 days with monocytes matured with h.i. *B. infantis*, and assayed for suppressive activity using CFSE-labeled resting target T cells and different ratios of T-cells stimulated prior assay onset for 3 days with *B. infantis*-matured monocytes. The mix of cells were then stimulated with anti-CD3 antibody in the presence of dendritic cells for 4 days, and proliferating cells were determined based on decrease in CFSE expression. Percent suppression was calculated based on CFSE dilution of target T-cells in suppression culture. Cumulative results are shown and each dot represents one donor. Correlation analyses was performed using Spearman's Rho, with *** representing $p < 0.001$. Data shown are representative of at least 5 experiments performed with T-cells from different donors. The lines show a linear approximation of the logarithmically transformed ratio of T suppressor cells to target cells.

Minor issues:

there are a number of stylistic and grammatical errors and some labelling issues: mislabeling in the legends/graphs, e.g. Fig 2 D & E appear switched?

We apologize for the switch of Fig. 2D and E and corrected it in the figure legend. We had to corrected unfortunately quite some minor issues in the legends.

Fig 3: isolated from

Corrected: CD25 expression of naïve T-cells form neonates, infants, children and adults in response to h.i. *B. longum*-matured monocytes for 3 days in the presence or absence of HLA-DR blocking antibody.

Fig 4: all of a sudden called corona patients?

We exchanged the expression into "in severely ill COVID-19 patients"

Reviewer #3 (Remarks to the Author):

Thank you for inviting me to review the manuscript. The authors describe the effects of several relevant bacteria on the responses and development of T-cells obtained from individuals across the lifespan.

The manuscript is well written and the topic is quite relevant.

We are pleased to read the statement of reviewer #3: "The manuscript is well written and the topic is quite relevant."

I have several comments that I hope will improve the manuscript. I have two major comments and some other minor comments that I believe the authors can easily address.

Major points:

1. Line 77-78: *B. infantis* is actually very rare among infants in the EU and USA (for example, see PMIDs: 27133167 and 33479326) and is enriched only in a limited number of communities (e.g., PMID:33905556). It would be more accurate to note that this organism is capable of colonizing the infant gut early in life (rather than noting that it 'is one of the earliest colonizers') (cite PMID:19033196) and outcompeting *Staphylococcus* and *Streptococcus* (cite PMID:29242832), decreases inflammation (cite PMID:34143954) and noting that its growth is supported by components found in human milk (instead of "provided by mother's milk") (PMID 21342711).

The citations noted don't fully support the (correct) statement that *B. infantis* out-competes other potentially pathogenic/inflammatory taxa, but adding the recommended citations above and adjusting the text will improve the accuracy of the statement.

We would like to thank the reviewer for the excellent refinement of our text. We have adjusted the text and added the citations.

2. Line 383: A major question I have about this work is how the cells were heat inactivated. It is not clear from the manuscript, but excessive heating might remove some of the volatile metabolites produced by the cells.

In order to exclude the possibility of ignoring metabolites due to heat inactivation, we carried out the inhibition experiments with living *B. infantis*. In fact, the inhibition induced by *B. infantis*-stimulated T cells is just as strong. Data are shown below and in the Fig. 5 of the manuscript.

Fig. for reviewer's use (shorter version within the manuscript): The suppressive capacity of h.i. (inactive) and living *B. infantis*-primed T cells is similar. T-cells were prepared and analyzed as in Fig. 4. Monocytes were matured with either live (living) or h.i. (inactive) *B. infantis* as indicated and naive neonatal or naive adult T cells were co-cultured with them. Orange squares highlight that in both set ups the *S. aureus*-specific response was strongly inhibited. Each dot represents a different donor. Error bars in the figures denote \pm SD. ** $p < 0.01$, *** $p < 0.001$, **** $p < 0.0001$ as determined by one-way ANOVA with Holm-Sidak post hoc test.

Further, were the cells washed before heat inactivation? Please carefully describe the methods for the generation of the cells used here.

Yes, the bacteria were extensively washed before heat inactivation and before adding the T cells to the monocytes. We added more information on this into Materials and Methods.

Minor comments:

1. The use of *B. longum* throughout is a little confusing (e.g., Line 77) when referring to *B. infantis*. While this is the species name, the subspecies is most commonly used to refer to the organism in a shortened form after the first usage. Given the phenotypic distinction between *B. longum* subsp. *longum* and *B. longum* subsp. *infantis*, it seems most appropriate to use *B. infantis* throughout.

For example, on the first usage use: "...Bifidobacterium longum subsp. *infantis* (*B. infantis*)..."

A very good point. We use *B. infantis* throughout, now. Thank you.

2. It is unclear from the manuscript what the number of donors was or any demographics

about them (e.g., age). It would be helpful to include this information, maybe as a table, to note the age and number (and other relevant characteristics) of the participants.

We apologize for missing information of participants. They all (or their parents) answer questionnaires and are well characterized. Three tables (Tab. S1-S2) of detailed information about the demographics were added to the supplemental material.

3. Line 202-219: I think the authors might mean genes as this is RNASeq data being described. Or, if it's a number of classes of compounds, list them in a table with their respective references?

...we clarified the issue in the text.

4. Line 227-228: "By analyzing..." is a sentence fragment.

We corrected the fragment.

5. Line 234: "parameters"

Corrected.

6. Line 251: include a comma after "early in life" for clarity

Done.

7. Line 278: It's unclear what the authors are referencing by "peptides pulsed monocytes to..."

We mean peptide loaded monocytes and clarified this in the text.

8. Line 351: The authors should soften this statement. As there is not an identified mechanism for the interactions observed with Bifidobacterium, it is premature to broadly reference the use of any Bifidobacterium as other strains or species may not act in the same way. Please soften this statement, refer specifically to *B. infantis*, and/or add a note cautioning that the mechanism for the observed functions is not known.

Thank you for this comment, we referred to *B. infantis*, now. In addition, we added the issue about the difference to other *B. longum* subspecies less and more careful.

9. Line 352: "corona patients" needs to be revised.

Was exchange to "COVID-19 patients" in the main body of the manuscript as well as in the Supplemental Figures.

REVIEWERS' COMMENTS

Reviewer #1 (Remarks to the Author):

I thank the authors for their careful and diligent responses to my comments. I greatly enjoyed reading the article, which I felt was able to convey the strength of the results as they pertain to early life human immunology. No further revisions are requested.